# DiCoFlex: Model-agnostic diverse counterfactuals with flexible control

**Oleksii Furman**[1]    **Ulvi Movsum-zada**[*,2,3]    **Patryk Marszalek**[*,2,3]    **Maciej Zięba**[1,4]

**Marek Śmieja**[2]

[1]Wrocław University of Science and Technology
[2]Faculty of Mathematics and Computer Science, Jagiellonian University, Kraków, Poland
[3]Doctoral School of Exact and Natural Sciences, Jagiellonian University, Kraków, Poland
[4]Tooploox Sp. z o.o.

oleksii.furman@pwr.edu.pl, ulvi.movsumzade@gmail.com,
patryk.marszalek@doctoral.uj.edu.pl, maciej.zieba@pwr.edu.pl
marek.smieja@uj.edu.pl

## Abstract

Counterfactual explanations play a pivotal role in explainable artificial intelligence (XAI) by offering intuitive, human-understandable alternatives that elucidate machine learning model decisions. Despite their significance, existing methods for generating counterfactuals often require constant access to the predictive model, involve computationally intensive optimization for each instance and lack the flexibility to adapt to new user-defined constraints without retraining. In this paper, we propose DiCoFlex, a novel model-agnostic, conditional generative framework that produces multiple diverse counterfactuals in a single forward pass. Leveraging conditional normalizing flows trained solely on labeled data, DiCoFlex addresses key limitations by enabling real-time user-driven customization of constraints such as sparsity and actionability at inference time. Extensive experiments on standard benchmark datasets show that DiCoFlex outperforms existing methods in terms of validity, diversity, proximity, and constraint adherence, making it a practical and scalable solution for counterfactual generation in sensitive decision-making domains.

## 1 Introduction

Counterfactual explanations (CFs) have become an integral part of explainable artificial intelligence (XAI) by providing human-interpretable insights into complex machine learning models [13]. CFs answer critical "what-if" questions by suggesting minimal and meaningful changes to input data that could alter the outcome of a predictive model [44]. Such explanations have valuable applications across sensitive domains, including finance, healthcare, and legal decisions, where understanding the predictions of the model and potential alternative outcomes is paramount [48, 11].

According to Guidotti [13], an ideal counterfactual explanation should satisfy several key properties. First, it must demonstrate **validity** by successfully changing the model's prediction. Second, it should maintain **proximity** by remaining as close as possible to the original input, minimizing the amount of change. Third, it should exhibit **sparsity** by modifying only the smallest possible set of

---

[*]Equal contribution.

39th Conference on Neural Information Processing Systems (NeurIPS 2025).

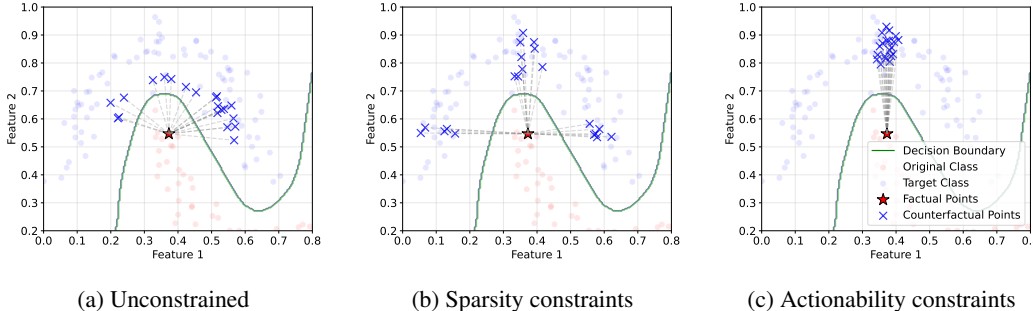

| (a) Unconstrained | (b) Sparsity constraints | (c) Actionability constraints |

Figure 1: Visualization of diverse counterfactual explanations generated by DiCoFlex for a single instance using an artificial two-class moon-shaped dataset. Each subfigure demonstrates different constraint scenarios: (a) no constraints applied, showing natural diversity; (b) sparsity constraints enforced through the $p$-norm parameter, resulting in minimal feature modifications; and (c) actionability constraints applied via feature masks, restricting which features can be modified.

features. Fourth, it must ensure **plausibility** by remaining within the realm of realistic, observed data. Finally, it should guarantee **actionability** by suggesting changes that are feasible for the stakeholder to implement.

In practice, however, these criteria often conflict. For instance, the smallest possible change may result in samples outside the realistic data distribution. To address this, Mothilal et al. [22] and Laugel et al. [20] argue that, rather than producing a single "best" counterfactual explanation, a *diverse* set of CFs should be generated. Presenting multiple realistic alternatives allows users to see different ways to achieve the desired outcome, explore trade-offs between minimal change and realism, and gain deeper insight into the model's decision boundaries. This diversity, in turn, empowers stakeholders to make more informed, feasible decisions.

Current approaches to diverse counterfactual generation face significant limitations. Optimization-based methods like DiCE [22] and multi-objective frameworks [7] incorporate diversity directly into their cost functions but require solving complex, separate optimizations for each explanation. This results in high computational costs and latency, making them impractical for real-time applications. Meanwhile, generative approaches such as those by Pawelczyk et al. [28] and Panagiotou et al. [25] offer efficiency improvements but frequently produce redundant explanations, require access to model gradients, and lack flexibility for adapting to user-defined constraints without extensive retraining.

To address these limitations, we introduce DiCoFlex, a novel conditional generative framework capable of generating multiple diverse CFs in a single forward pass. DiCoFlex is model-agnostic and does not require constant access to the underlying predictive model at any stage, relying solely on a dataset labeled by a classification model. Our approach leverages conditional normalizing flows to learn the distribution of counterfactual explanations that inherently satisfy user-specified constraints.

In contrast to existing methods, DiCoFlex incorporates customizable constraints such as sparsity and actionability directly at inference time, eliminating the need for retraining when constraints change. During inference, DiCoFlex achieves computational efficiency by generating multiple diverse CFs through a single forward pass, thus avoiding the computational burden inherent in methods that require separate optimization procedures for each counterfactual. As illustrated in Figure 1, our method can operate under different constraints. DiCoFlex generates diverse and plausible counterfactual explanations without any restrictions imposed on sparsity or actionability (as shown in Figure 1a). Sparsity, controlled by the user-defined exponent $p$ in $L_p$ norm, ensures minimal modifications in input features (as demonstrated in Figure 1b), while actionability constraints allow users to specify immutable features through a mask, guiding the generation towards practically feasible explanations (shown in Figure 1c). These parameters can be dynamically adjusted at inference time, providing unprecedented flexibility in generating counterfactual explanations that satisfy domain-specific constraints while maintaining a balance between diversity and validity without requiring explicit diversity penalties.

The main contributions of our work are summarized as follows:

- We propose DiCoFlex, a classifier-agnostic, conditional generative framework that generates multiple counterfactuals efficiently in a single forward pass, without the need for model retraining or optimization per query instance.
- Our method enables real-time customization of constraints including sparsity and actionability, providing practical, user-centric control over counterfactual generation.
- We extensively evaluate DiCoFlex on benchmark datasets, demonstrating superior performance in terms of validity, proximity, diversity, and constraint compliance compared to existing state-of-the-art approaches.

## 2 Related Work

Counterfactual explanations (CFs) have become integral to explainable AI by providing intuitive alternatives that elucidate model decisions [44]. Optimization-based methods formulate CF generation as minimizing distance between instances while ensuring prediction changes, using implementations including gradient-based approaches [44], integer programming [41], genetic algorithms [7], and satisfiability problems [18]. Instance-based methods like FACE [29] select examples yielding desired predictions from existing data, while CBCE [19] merges features from paired contrasting instances. Model-specific approaches such as decision tree methods [14, 38] leverage model structure by traversing alternative paths.

**Diverse Counterfactual Explanations**   Generating multiple diverse counterfactuals offers significant advantages over single explanations by providing alternative paths for recourse and better illustrating decision boundaries [20, 22]. DiCE [22] directly incorporates diversity into its loss function but requires separate optimization for each explanation, increasing computational costs. Other approaches achieve diversity through various strategies: DiVE [33] employs determinantal point processes, while methods like [39, 35, 5] partition the feature space. Multi-objective approaches like MOC [7] generate solutions representing different trade-offs between properties such as proximity and sparsity, while CARE [30] and OrdCE [17] incorporate user preferences into the generation process.

**Generative Models for Counterfactuals**   Generative models have gained popularity for counterfactual generation due to their ability to produce more plausible explanations that remain on the data manifold. VAE-based approaches include ReViSE [16], C-CHVAE [28], CLUE [2], and CRUDS [10], which learn latent representations of the data distribution. GAN-based methods such as GeCo [36] and SCGAN [48] use adversarial training to balance validity and realism. Recent transformer-based approaches like TABCF [25] specifically target tabular data with mixed types. Normalizing flows have also been explored for counterfactual generation [46, 12], where they model the class-conditional distribution for plausibility assessment during gradient-based optimization. However, many of these approaches still require access to model gradients, produce redundant explanations, or lack flexibility in handling user-defined constraints.

**Handling Constraints in Counterfactual Generation**   Practical counterfactual explanations must satisfy various constraints including sparsity (minimizing modified features) and actionability (ensuring feasible changes). CERTIFAI [37] employs genetic algorithms to enforce constraints, while MCCE [31] conditions on immutable features. Methods focused on actionable recourse, such as CSCF [23], account for downstream effects of feature modifications, while Russell [34] and Verma et al. [43] generate explanations representing different modification strategies. However, most existing approaches require retraining or reoptimization when constraints change, limiting their flexibility in real-world applications.

**Limitations of Current Approaches**   Current counterfactual methods face substantial limitations for tabular data: optimization-based approaches require high computational costs for diverse explanations [22], and generative models often produce redundant explanations, require model gradients, or lack constraint flexibility [28]. Our proposed approach, DiCoFlex, addresses these limitations through a conditional generative framework that efficiently produces diverse counterfactuals in a single forward pass, enables real-time constraint customization, operates without requiring model access, and naturally handles mixed feature types.

# 3 DiCoFlex Method

Given a classification model $h$ and an initial input example $\mathbf{x}_0$ within a $d$-dimensional real space $\mathbb{R}^d$, one seeks to determine a counterfactual instance $\mathbf{x}' \in \mathbb{R}^d$. A counterfactual $\mathbf{x}'$ represents a sample, which (i) is classified to desired class $y'$ by the model $h$ (i.e. $h(\mathbf{x}') = y'$), (ii) lies close enough to $\mathbf{x}_0$ according to some distance measure, $d(\mathbf{x}_0, \mathbf{x}')$, and (iii) is plausible (in-distribution sample).

This problem formulation focuses on the deterministic extraction of counterfactuals, where, for a given example $\mathbf{x}_0$ and desired class $y'$, the model returns a single counterfactual $\mathbf{x}'$. In practice, generating more plausible candidates provides a wider spectrum of possible explanations satisfying validity, plausibility, and proximity constraints. Therefore, we postulate an approximate conditional distribution for counterfactuals for a given example $\mathbf{x}_0$ and the target class $p(\mathbf{x}'|\mathbf{x}_0, y')$.

In this section, we first introduce a general objective function for a model generating counterfactual explanations. Next, we parametrize our model with normalizing flow and explain our strategy for scoring counterfactual candidates in a training phase. Finally, we introduce sparsity and actionability constraints and summarize the training algorithm.

## 3.1 The model objective

We consider the parameterized model $p_\theta(\mathbf{x}'|\mathbf{x}_0, y')$ for the distribution approximation. Direct optimization of $p_\theta(\mathbf{x}'|\mathbf{x}_0, y')$ is challenging, because we do not have access to the training ground-truth pairs composed of initial examples $\mathbf{x}_0$ and the corresponding counterfactuals $\mathbf{x}'$ for class $y'$. Therefore, we introduce conditional distribution $q(\mathbf{x}'|\mathbf{x}_0, y')$ and we utilize the expected value of Kullback-Leibler divergence (KLD) between $p_\theta(\mathbf{x}'|\mathbf{x}_0, y')$ and $q(\mathbf{x}'|\mathbf{x}_0, y')$ as training objective:

$$\mathcal{L} = \mathbb{E}_{\mathbf{x} \sim p_{\text{data}}, y' \sim \boldsymbol{\pi} \setminus \{h(\mathbf{x})\}} \left[ D_{\text{KL}}(q(\mathbf{x}'|\mathbf{x}, y') \parallel p_\theta(\mathbf{x}'|\mathbf{x}, y')) \right] \tag{1}$$

$$= \mathbb{E}_{\mathbf{x}, y'} \mathbb{E}_{\mathbf{x}' \sim q(\mathbf{x}'|\mathbf{x}, y')} \left[ \log \frac{q(\mathbf{x}'|\mathbf{x}, y')}{p_\theta(\mathbf{x}'|\mathbf{x}, y')} \right] \tag{2}$$

$$= \mathbb{E}_{\mathbf{x}, y'} \mathbb{E}_{\mathbf{x}' \sim q(\mathbf{x}'|\mathbf{x}, y')} \left[ \log q(\mathbf{x}'|\mathbf{x}, y') - \log p_\theta(\mathbf{x}'|\mathbf{x}, y') \right], \tag{3}$$

where $\mathbf{x}$ is an example from the data distribution $p_{\text{data}}$ (training example) and $y'$ is sampled from categorical distribution $\boldsymbol{\pi}$ representing the prior over classes, excluding the current class of $\mathbf{x}$, predicted by $h(\cdot)$. The component $q(\mathbf{x}'|\mathbf{x}, y')$ does not depend on parameters $\theta$, therefore the final objective is:

$$\mathcal{Q} = -\mathbb{E}_{\mathbf{x}, y'} \mathbb{E}_{\mathbf{x}' \sim q(\mathbf{x}'|\mathbf{x}, y')} \left[ \log p_\theta(\mathbf{x}'|\mathbf{x}, y') \right]. \tag{4}$$

We further elaborate on how $p_\theta(\mathbf{x}'|\mathbf{x}, y')$ is modeled and how sample from $q(\mathbf{x}'|\mathbf{x}, y')$.

## 3.2 Modeling predictive conditional distribution

Our approach aims at directly optimizing the conditional negative log-likelihood given by (4) with the data sampled from the distribution $q(\mathbf{x}'|\mathbf{x}, y')$. As a consequence, it requires access to the parametrized density function. Several techniques, such as Kernel Density Estimation (KDE) and Gaussian Mixture Models (GMM), can be used to model the conditional density function $p_\theta(\mathbf{x}'|\mathbf{x}, y')$. In this work, we propose using a conditional normalizing flow model [32] for this purpose. Unlike KDE or GMM, normalizing flows do not rely on a predefined parametric form of the density function and are well-suited for modeling high-dimensional data. Furthermore, in contrast to other generative models, such as diffusion models or GANs, normalizing flows allow for exact density evaluation via the change of variables formula and can be efficiently trained by minimizing the negative log-likelihood. The general formula that represents normalizing flows can be expressed as:

$$p_\theta(\mathbf{x}'|\mathbf{x}, y') = p_Z(f_\theta(\mathbf{x}'; \mathbf{x}, y')) \cdot |\det J_{f_\theta}(\mathbf{x}'; \mathbf{x}, y')|, \tag{5}$$

where $f_\theta(\mathbf{x}'; \mathbf{x}, y')$ is an invertible transformation conditioned on $\mathbf{x}$ and $y'$, $p_Z(f_\theta(\mathbf{x}'; \mathbf{x}, y'))$ is the base distribution, typically Gaussian and $J_{f_\theta}(\mathbf{x}'; \mathbf{x}, y')$ is Jacobian of the transformation. The biggest

challenge in normalizing flows is the choice of the invertible function for which the determinant of Jacobian is easy to calculate. Several solutions have been proposed in the literature to address this issue with notable approaches, including NICE [8], RealNVP [9], and MAF [26]. Based on superior performance on benchmark dataset, we selected MAF as our normalizing flow model (see Appendix D).

### 3.3 Selecting training counterfactual explanations

The distribution $q(\mathbf{x}'|\mathbf{x}, y')$ is essential to sample data to train the flow model. Since we do not have direct access to this distribution, we introduce the straightforward technique of querying training counterfactual examples $\mathbf{x}'$, which are similar to the input data and satisfy $h(\mathbf{x}') = y'$. For this purpose, we use a typical KNN (K-nearest neighbors) heuristic, in which we sample $K$ data points labeled as $y'$ by the classifier that are closest to the input $\mathbf{x}$. Since we are sampling *training examples* from *class $y'$*, the validity and plausibility constraints hold, while the proximity is satisfied by using the *KNN approach*.

Concluding, the approximation for the probability distribution $q(\cdot)$ is as follows:

$$\hat{q}(\mathbf{x}'|\mathbf{x}, y', d) = \begin{cases} \frac{1}{K} & \text{if } \mathbf{x}' \in N(\mathbf{x}, y', d, K), \\ 0 & \text{otherwise,} \end{cases} \tag{6}$$

where $N(\mathbf{x}, y', K, d)$ is set of $K$ closest neighbors of $\mathbf{x}$ in class $y'$, considering distance measure $d(\cdot, \cdot)$.

One could ask whether a direct use of the distribution $\hat{q}(\mathbf{x}' \mid \mathbf{x}, y', d)$ can be an alternative for generating counterfactual explanations, instead of learning a parametrized flow-based model $p_\theta(\mathbf{x}' \mid \mathbf{x}, y')$. Observe that the use of $\hat{q}$ is inherently limited by the number of available training examples and lacks the ability to generalize to unseen instances. Additionally, the diversity of generated examples is constrained by the number of selected neighbors. Modeling a generative model provides smooth changes over input examples and allows for extrapolating counterfactual generation outside training data. Importantly, our sampling strategy provides theoretical guarantees by construction: sampled counterfactuals are guaranteed to be valid, proximal, and plausible. We formalize these properties in Propositions A.3, A.4, and A.5 (Appendix A).

### 3.4 Sparsity on continuous features

In many practical use cases, counterfactuals need to satisfy additional constraints, such as sparsity. This means that, in addition to minimizing the Euclidean distance, a counterfactual has to minimize the number of modified attributes. While sparsity is often implicitly achieved for categorical variables due to the applied optimization scheme, enforcing sparsity on continuous features remains challenging. We show that the sparsification of continuous attributes can be controlled in the inference phase by an auxiliary conditioning factor in a flow model.

Instead of querying neighbor examples to $x$ according to Euclidean distance in KNN search, we use $L_p$ distance function given by:

$$d_p(\mathbf{x}'_i, \mathbf{x}) = ||\mathbf{x}'_i - \mathbf{x}||_p = \left( \sum_{j=1}^{D} |x'_j - x_j|^p \right)^{1/p}. \tag{7}$$

We can observe that the level of modified attributes can be controlled through an appropriate choice of the value of $p$. For instance, if $p = 0.01$, neighbors with fewer modified attributes are favored, thereby enforcing sparsity. Making use of $p = 2$ gives a standard Euclidean proximity measure. Since categorical attributes are commonly encoded by one-hot vectors, the $L_p$ distance exhibits a discrete jump property: the distance equals 0 if categories match and a fixed value otherwise, regardless of $p$. This discrete structure naturally limits categorical changes. Therefore, the above scheme affects mostly continuous features.

In practice, the sparsity level can be effectively controlled in the inference time by incorporating the parameter $p$ into the flow model $p_\theta(\mathbf{x}' \mid \mathbf{x}, y', p)$ through additional conditioning. During training, values of $p$ are sampled and explicitly included in the conditioning of the model, allowing it to learn

the relationship between $p$ and the desired sparsity in the generated outputs. As a result, during the inference stage, users can adjust the sparsity level of the modified attributes simply by selecting an appropriate value of $p$. This mechanism provides a flexible and interpretable way to regulate the extent of changes applied to the input $\mathbf{x}$, depending on specific needs or constraints. Note, however, that the applied $L_p$ norm does not freeze the continuous attributes but only encourages small changes on their portion, which can be seen as a type of soft constraint. From a theoretical perspective, using small $p$ values (e.g., $p = 0.01$) restricts the neighborhood $N(x, y', d_p, K)$ to a more concentrated region of the feature space, which typically corresponds to higher-density areas of the data manifold. This explains why DiCoFlex achieves improved plausibility scores empirically, as formalized in Proposition A.5.

### 3.5 Actionability constraints

One of the crucial aspects of counterfactual explanations is enforcing the manipulation of some particular attributes. In order to accomplish that, in our approach, we introduce a further modification in a distance function while calculating the nearest neighbors:

$$d_{p,\mathbf{m}}(\mathbf{x}, \mathbf{x}') = \|\mathbf{x} - \mathbf{x}'\|_{p,m} = \alpha \sum_{j=1}^{D} m_j |x_j - x_j'|^p + \sum_{j=1}^{D} (1 - m_j)|x_j - x_j'|^p, \qquad (8)$$

where $\mathbf{m} \in \{0, 1\}^D$ is binary vector, for which $m_d = 1$ indicates the attributes that should not be modified. The value of hyperparameter $\alpha$ should be large enough to prevent attributes $m_d = 1$ from modification.

During training the masking vectors $\mathbf{m}$ are sampled from some set of possible mask patterns $\mathcal{M}$ and the mask is also delivered as a conditioning factor to the flow-based model, which represents $p_\theta(\mathbf{x}' \mid \mathbf{x}, y', p, \mathbf{m})$. Once trained with a set of masks $\mathcal{M}$, DiCoFlex can dynamically change the mask during inference resulting in imposing different user expectations. Similar to enforcing sparsity, here we do not prevent from modifying masked attributes completely but only encourage the model to prioritize changing unmasked features first.

### 3.6 Training

Concluding, the training procedure for estimating $\theta$ is as follows. For each training iteration, $\mathbf{x}$ is selected from data examples and the target class $y'$ is sampled for counterfactual explanation. Value $p$ that represents the sparsity level is also sampled from some predefined set $\mathcal{P}$, as well as the masking vector $\mathbf{m}$ from $\mathcal{M}$. Next, the set $N(\mathbf{x}, y', d_{p,\mathbf{m}}, K)$ of $K$ nearest neighbors is calculated considering distance measure $d_{p,\mathbf{m}}$ given by (8). The counterfactual $\mathbf{x}'$ is sampled from $\hat{q}(\mathbf{x}'|\mathbf{x}, y', d_{p,\mathbf{m}})$ given by (6). Finally, $\theta$ is updated in gradient-based procedure by optimizing (4), considering conditioning both on $p$ and $\mathbf{m}$. Appendix E contains the details of the training algorithm.

The training procedure inherits several desirable theoretical properties from the $K$ nearest neighbors sampling mechanism. First, all training samples are guaranteed to be valid counterfactuals by construction (Proposition A.3). Second, proximity is controlled through an implicit upper bound determined by the $K$-th nearest neighbor (Proposition A.4). Third, plausibility is ensured as all samples originate from the training distribution (Proposition A.5). Finally, the normalizing flow learns to preserve diversity with a formal guarantee that expected diversity remains close to the empirical diversity, with deviation bounded by the training error (Theorem A.2). These theoretical foundations provide principled justification for the empirical performance demonstrated in Section 4.

## 4 Experiments

In this section, we conduct a series of experiments that evaluate the quality of counterfactual explanations generated by various methods[2]. Our main objective is to understand how effectively these counterfactuals flip the predictions of a model while maintaining realism, requiring minimal changes to the input, and offering diversity among the generated alternatives.

---

[2]Code available at `https://github.com/ofurman/DiCoFlex`

### 4.1 Experimental Setup

**Datasets**   We evaluate all methods on five well-established benchmark datasets commonly used in counterfactual explanation research: Adult [3] (income prediction), Bank Marketing [21] (customer response classification), Default [47] (credit card default risk prediction), Give Me Some Credit (GMC) [28] (financial insolvency prediction), and Lending Club [15] (loan creditworthiness classification), see Appendix B for details. These datasets are selected for their diversity in dimensionality, feature types (mixed categorical and continuous variables), and real-world applicability, allowing us to evaluate the robustness and generalizability of our approach across different domains.

**Parameterization of DiCoFlex**   For our experimental evaluation, we compare two variants of our method: DiCoFlex with default Euclidean distance (p=2.0) and DiCoFlex (p=0.01), which uses the $L_p$ norm with $p$=0.01. This latter version encourages higher sparsity for continuous features, leading to explanations that alter fewer continuous attributes while maintaining validity, see Section 3.5. Importantly, these configurations are not separate methods but represent different inference-time settings of the same trained model. This design demonstrates DiCoFlex's ability to dynamically adjust sparsity and proximity constraints without any retraining, enabling continuous user control over counterfactual behavior.

**Baseline Methods**   We compare our approach against several state-of-the-art counterfactual explanation methods. First of all, we consider methods, which generate multiple explanations for a single data point. CCHVAE [28] employs a conditional variational autoencoder to generate counterfactuals consistent with the data distribution. DiCE [22] uses gradient-based optimization to produce diverse counterfactuals in a model-agnostic framework. For all these methods, we generate up to 100 counterfactual explanations for each factual instance whenever feasible and randomly select 10 samples based solely on the validity criterion to provide an equal number of counterfactuals for every method.

Additionally, we compare with methods focused on returning a single explanation. ReViSE [16] introduces a probabilistic framework designed to separate sensitive and non-sensitive features with a focus on interpretability. The approach by Wachter et al. [44] represents a pioneering optimization-based method that minimizes a loss function balancing proximity to the input with the goal of changing the prediction. TABCF [25] specifically targets tabular data with a transformer-based VAE approach. This class of methods is expected to perform better because they are optimized for producing a single best explanation while the first class of methods takes a diversity criterion as a priority.

**Evaluation Metrics**   We evaluate counterfactual quality using metrics aligned with key desirable properties. For test instances originally classified as class 0, we generate corresponding counterfactuals targeting class 1 and vice versa. **Validity** measures the percentage of counterfactuals that successfully change model predictions to the target class. **Hypervolume** [24] is used to measure diversity to quantify the spread and coverage of counterfactuals in the objective space. Plausibility is assessed through **Local Outlier Factor (LOF)** [4] that measure local density deviation of a data point compared to its neighbors. **Sparsity** quantifies feature modifications, calculated as the proportion of changed features in each category. For continuous features, we use $\varepsilon$**-sparsity**, which counts features modified beyond a relative threshold of 5% of the original feature value. **Proximity** measures distance between original instances and the counterfactuals using the Manhattan distance only for continuous features. Observe that for categorical features proximity coincides with the sparsity. **Probability** measures model confidence in the target prediction. For the sparsity, proximity and plausibility metrics, lower values indicate better performance ($\downarrow$), while for the validity, probability and diversity metrics, higher values are preferable ($\uparrow$). We provide a detailed description of the metrics in Appendix C.

### 4.2 Evaluating multiple explanations

The comparison with baselines for diverse counterfactual explanations (DiCE, CCHVAE) presented in Table 1 confirms that DiCoFlex returns more diverse explanations than competitive methods, achieving superior hypervolume scores. In terms of plausibility evaluation, both versions of DiCoFlex consistently outperform other baseline methods as indicated by the LOF measure (except the GMC dataset). Furthermore, the results reveal that the proposed method is capable of producing counterfactual explanations with a low proximity score on continuous attributes.

Table 1: Comparison of counterfactual generation methods across datasets. The best method, which generates multiple counterfactual candidates for each metric is highlighted in **bold** (DiCoFlex, CCHVAE, DiCE). If there is a method that generates a single counterfactual (ReViSE, TABCF, Wachter) that is better than the bold result, it is underlined.

| Dataset | Model | Validity. ↑ | Classif. prob. ↑ | Proximity cont. ↓ | Sparsity cat. ↓ | ε-sparsity cont. ↓ | LOF log scale ↓ | Hypervol. log scale ↑ |
|---|---|---|---|---|---|---|---|---|
| Lending Club | DiCoFlex (p=2.0) | **1.000** | **0.999** | **0.668** | 0.412 | 0.840 | 0.476 | **13.118** |
| | DiCoFlex (p=0.01) | **1.000** | 0.997 | 0.704 | 0.635 | **0.737** | **0.453** | 11.373 |
| | CCHVAE | **1.000** | 0.682 | 0.769 | **0.137** | 0.895 | 0.519 | 7.448 |
| | DiCE | **1.000** | 0.978 | 3.444 | 0.146 | 0.840 | 0.567 | 8.252 |
| | ReViSE | 0.910 | 0.721 | 0.588 | 0.077 | 0.639 | 0.040 | - |
| | TABCF | **1.000** | 0.946 | 0.557 | 0.635 | 0.551 | 0.047 | - |
| | Wachter | 0.950 | 0.530 | 1.268 | 0.000 | 0.696 | 0.943 | - |
| Adult | DiCoFlex (p=2.0) | **1.000** | 0.998 | 0.581 | 0.515 | 0.498 | 2.156 | **2.036** |
| | DiCoFlex (p=0.01) | **1.000** | **1.000** | **0.373** | 0.568 | **0.350** | **1.812** | 0.379 |
| | CCHVAE | **1.000** | 0.894 | 0.616 | **0.302** | 0.582 | 2.176 | 1.571 |
| | DiCE | **1.000** | **1.000** | 5.096 | 0.557 | 0.582 | 3.815 | 0.681 |
| | ReViSE | 0.860 | 0.879 | 1.004 | 0.206 | 0.576 | 2.637 | - |
| | TABCF | **1.000** | 0.986 | 3.129 | 0.263 | 0.348 | 3.746 | - |
| | Wachter | 0.970 | 0.578 | 1.046 | 0.000 | 0.830 | 2.893 | - |
| Bank | DiCoFlex (p=2.0) | **1.000** | 0.971 | 0.717 | 0.499 | 0.786 | 0.276 | **3.808** |
| | DiCoFlex (p=0.01) | **1.000** | 0.894 | **0.584** | 0.480 | **0.673** | **0.247** | 3.006 |
| | CCHVAE | **1.000** | 0.951 | 0.989 | 0.357 | 0.816 | 0.778 | 0.309 |
| | DiCE | **1.000** | **0.994** | 7.117 | **0.106** | 0.787 | 0.839 | 0.323 |
| | ReViSE | 0.230 | 0.823 | 1.119 | 0.285 | 0.615 | 0.320 | - |
| | TABCF | **1.000** | 0.850 | 3.747 | 0.210 | 0.441 | 0.360 | - |
| | Wachter | 0.830 | 0.530 | 3.826 | 0.000 | 0.962 | 0.887 | - |
| Default | DiCoFlex (p=2.0) | **1.000** | 0.986 | **0.459** | 0.419 | 0.831 | 0.038 | **62.427** |
| | DiCoFlex (p=0.01) | **1.000** | **0.988** | 0.489 | 0.514 | 0.787 | **0.034** | 61.593 |
| | CCHVAE | **1.000** | 0.963 | 0.679 | 0.463 | 0.832 | 0.047 | 53.706 |
| | DiCE | **1.000** | 0.914 | 1.039 | **0.263** | **0.692** | 0.373 | 51.184 |
| | ReViSE | 0.170 | 0.784 | 0.595 | 0.281 | 0.613 | 0.033 | - |
| | TABCF | **1.000** | 0.884 | 0.516 | 0.510 | 0.412 | 0.057 | - |
| | Wachter | 0.930 | 0.557 | 4.494 | 0.000 | 0.976 | 0.658 | - |
| GMC | DiCoFlex (p=2.0) | **1.000** | **0.969** | 0.690 | 0.755 | 0.765 | 0.861 | 15.667 |
| | DiCoFlex (p=0.01) | **1.000** | 0.968 | 0.781 | 0.764 | 0.763 | 0.899 | **15.810** |
| | CCHVAE | **1.000** | 0.932 | **0.389** | 0.756 | 0.762 | 0.564 | 9.761 |
| | DiCE | **1.000** | 0.934 | 3.478 | **0.753** | **0.699** | **0.399** | 9.184 |
| | ReViSE | 0.780 | 0.909 | 8.525 | 0.180 | 0.877 | 0.702 | - |
| | TABCF | 0.130 | 0.630 | 0.521 | 0.359 | 0.835 | 0.303 | - |
| | Wachter | 0.950 | 0.584 | 8.588 | 0.000 | 0.987 | 2.436 | - |

However, impressive diversity, plausibility, and proximity come at the price of slightly worse sparsity. Although DiCoFlex (p=0.01) obtained the lowest $\epsilon$-sparsity on continuous features in three out of five cases, the categorical sparsity is higher than the one returned by DiCE and CCHAVE. It is apparent that these two methods switch categorical variables less often than continuous ones, which is usually caused by the applied optimization scheme and can be seen as their drawback. Observe that the use of DiCoFlex (p=0.01) allows us to balance the level of sparsity between categorical and continuous features.

Our analysis confirms that typical measures for evaluating counterfactual explanations are in conflict and that there does not exist a method that optimizes all these criteria simultaneously. However, the proposed method compares favorably with other approaches beating them in diversity, plausibility, and proximity on continuous features.

### 4.3 DiCoFlex (p=2.0) vs. DiCoFlex (p=0.01)

As mentioned, lowering the value of $p$ in the $L_p$ norm for DiCoFlex directly results in lowering the sparsity measure for continuous attributes. As a side effect, it allows for larger changes in values of categorical variables to produce valid counterfactuals. An interesting consequence is that DiCoFlex (p=0.01) generates more plausible samples than DiCoFlex (p=2.0) (see the LOF score), which may be attributed to the way we model the distribution $q$ used for scoring counterfactual candidates; see

Section 3.3. For low $p = 0.01$, we restrict the support of $q$ to a small region of data space, where only a small number of coordinates are likely to be modified. Increasing this value leads to neighbors located in a larger and more diverse region since we allow for modifying all coordinates without extra penalty. Consequently, DiCoFlex with $p = 2$ can sample out-of-distribution counterfactuals more often than DiCoFlex (p=0.01) with $p = 0.01$. In practice, dynamic change of $p$ in inference allows users to examine multiple counterfactual candidates and select the one that suits their expectations best.

### 4.4  Comparison with single counterfactual methods

To present a broader view on the performance of DiCoFlex, we include an additional comparison with commonly used methods that generate a single explanation (ReViSE, TABCF, Wachter) (see Table 1). We note that DiCoFlex and DiCoFlex (p=0.01) still demonstrate impressive performance in terms of LOF score in this comparison. They both achieve the lowest LOF on Adult and Bank datasets and are comparable to the best-performing ReViSE method on the Default dataset. Moreover, they obtained the best continuous proximity measure in 3 out of 5 cases. This means that even though DiCoFlex produces multiple diverse counterfactuals for each instance, most of them satisfy the highest requirements used in counterfactual analysis.

Among competitive methods, ReViSE and TabCF represent strong baselines that achieve a satisfactory level of plausibility and sparsity. Nevertheless, these models often fail to generate valid counterfactuals (see TABCF on GMC or ReViSE on Bank and Default), which partially limits their usefulness. Generating multiple diverse counterfactuals (as in our method) allows the user to select suitable candidates that meet the expected criteria. It is interesting that TABCF and ReViSE obtain more than 10 times lower LOF on the Lending Club dataset than other approaches. It might follow from the fact that it is impossible to generate multiple good counterfactuals for this dataset, and only a few nearby candidates represent in-distribution samples. We also note that the classical Wachter method does not change categorical variables at all and generally gives suboptimal results, except for the validity score.

### 4.5  Runtime of counterfactuals generation

A key advantage of our approach is that once trained, Di-CoFlex can be used to generate multiple counterfactual explanations in a single forward pass. To illustrate this gain, we measure the time every method needs to generate a single counterfactual for 100 data points. Figure 2 confirms that both variants of DiCoFlex achieve generation times of $0.12 \pm 0.05$ seconds, outperforming other baseline methods by orders of magnitude. Optimization-based methods demonstrate substantially higher computational costs, with DiCE, ReViSE, and TABCF all requiring over 1000 seconds on average. This efficiency enables real-time exploration of diverse counterfactual explanations with dynamic constraint adjustment, addressing a critical limitation of existing approaches.

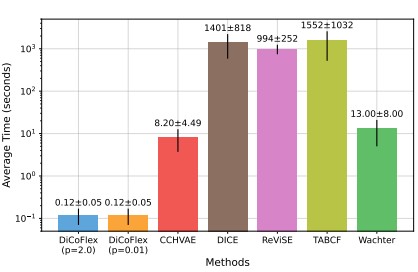

Figure 2: Average runtime (log scale).

### 4.6  Actionability Control

DiCoFlex is capable of imposing additional constraints when generating counterfactuals without retraining the model. As demonstrated in Section 4.3 and Table 1, the use of the $L_p$ norm with small $p$ allows to minimize the modifications of the continuous features and improve plausibility. In this experiment, we show the effect of introducing actionability constraints, which prevents severe modification of user-defined features.

For illustration, we train DiCoFlex on the Adult dataset with four mask configurations, which cover one or two features, as shown in Table 2. For instance, the use of "mask 1" should block changes of "Capital Gain" and "Capital Loss" attributes. As can be observed in the result, the application of every individual mask indeed decreases the sparsity level compared to the unconstrained version when no mask is used. In consequence, this flexible control enables users to dynamically adjust both sparsity

Table 2: Comparison of the unconstrained version of DiCoFlex with its masked variant, evaluated on features selected by the given mask. We report $\epsilon$-sparsity for numerical variables and standard sparsity for categorical ones (lower score means less attributes affected).

| | mask 1 | | mask 2 | mask 3 | | mask 4 | |
| Method | Capital Gain | Capital Loss | Age | Race | Sex | Native Country |
|---|---|---|---|---|---|---|
| Masked-DiCoFlex | 0.023 | 0.037 | 0.069 | 0.052 | 0.045 | 0.030 |
| Standard DiCoFlex | 0.148 | 0.097 | 0.833 | 0.170 | 0.200 | 0.116 |

and actionability, generating counterfactuals satisfying different domain-specific requirements without requiring model retraining. Appendix I contains additional evaluation of these models.

### 4.7 Sensitivity Analysis of Sparsity Parameter $p$

We further analyze the sensitivity of DiCoFlex to the sparsity control parameter $p$ that governs the $L_p$ norm used during neighbor selection (Section 3.4). This parameter determines the strength of sparsity constraints applied to continuous features and thus influences the trade-off between proximity, sparsity, and diversity.

Table 3 reports the performance of DiCoFlex across several $p$ values on the *Adult* dataset. Lower values (e.g., $p = 0.01$) favor sparse modifications, yielding counterfactuals that stay closer to the original instances, while larger values (e.g., $p = 1$ or $2$) permit more flexible changes that improve diversity at the cost of proximity.

Table 3: Effect of sparsity parameter $p$ on DiCoFlex performance on the *Adult* dataset. Lower proximity and LOF, higher validity and hypervolume indicate better performance.

| Model | Classif. prob. ↑ | Proximity cont. ↓ | Sparsity cat. ↓ | $\epsilon$-sparsity cont. ↓ | LOF log ↓ | Hypervol. log ↑ |
|---|---|---|---|---|---|---|
| DiCoFlex (p=0.01) | **1.000** | **0.373** | 0.568 | **0.441** | **1.9119** | 0.9023 |
| DiCoFlex (p=0.08) | 0.996 | 0.501 | 0.553 | 0.454 | 1.9680 | 1.0801 |
| DiCoFlex (p=0.25) | 0.995 | 0.551 | 0.541 | 0.501 | 2.2380 | 1.3780 |
| DiCoFlex (p=1.0) | 0.996 | 0.577 | 0.537 | 0.589 | 2.6349 | **1.9491** |
| DiCoFlex (p=2.0) | 0.998 | 0.581 | **0.515** | 0.601 | 2.6614 | 1.8862 |

Across all $p$ values, DiCoFlex maintains perfect validity, confirming robustness of the learned conditional flow. As $p$ increases, proximity gradually worsens while diversity (hypervolume) improves, reflecting a smooth proximity–sparsity trade-off. The consistent trend in LOF indicates stable plausibility across regimes.

These findings demonstrate that a single DiCoFlex model, trained with conditioning on $p$, can smoothly adapt its behavior to user preferences at inference time—ranging from highly sparse to diverse counterfactuals—without retraining or model modification.

## 5 Conclusion

We presented DiCoFlex, a model-agnostic conditional generative framework for counterfactual explanations that addresses key limitations in existing approaches. By leveraging normalizing flows, our method enables counterfactual generation without constant access to the classification model, provides inference-time control over constraints, and efficiently produces multiple diverse explanations in a single forward pass. Our approach is grounded in rigorous theoretical foundations: we prove formal guarantees for validity, proximity bounds, in-distribution plausibility, and diversity preservation (Appendix A). Empirical evaluations on five benchmark datasets demonstrate that DiCoFlex outperforms the state-of-the-art methods in diversity and validity while achieving superior plausibility and computational efficiency. Although optimization-based methods may achieve better sparsity in some cases, our approach offers a more balanced trade-off between competing explanation criteria.

## Acknowledgments and Disclosure of Funding

Oleksii Furman and Maciej Zieba's work was supported by the National Science Centre (Poland) Grant No. 2024/55/B/ST6/02100. The research of U. Movsum-zada and P. Marszałek was supported by the National Science Centre (Poland), grant no. 2023/50/E/ST6/00169. The work of M. Śmieja was supported by the National Science Centre (Poland), grant no. 2022/45/B/ST6/01117. Some experiments were performed on servers purchased with funds from the flagship project entitled "Artificial Intelligence Computing Center Core Facility" from the DigiWorld Priority Research Area within the Excellence Initiative – Research University program at Jagiellonian University in Krakow.

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

# A Theoretical Considerations

This section provides theoretical analysis of the diversity guarantees provided by DiCoFlex. We establish a formal lower bound on the expected diversity of counterfactuals generated by our learned conditional normalizing flow.

## A.1 Preliminary Lemma

**Lemma A.1** (Upper bound on difference of expected values). *Let $p$ and $q$ be two distributions over the same finite set $\mathcal{X}$, and let the total variation distance be defined as $\|p-q\|_{TV} = \frac{1}{2} \sum_{x \in \mathcal{X}} |p(x) - q(x)|$. For any function $f : \mathcal{X} \to \mathbb{R}$, we have:*

$$\left| \sum_{x \in \mathcal{X}} f(x)(p(x) - q(x)) \right| \leq 2 \cdot \max_{x \in \mathcal{X}} |f(x)| \cdot \|p - q\|_{TV}. \tag{9}$$

*Proof.* By the triangle inequality and the definition of total variation distance:

$$\left| \sum_{x \in \mathcal{X}} f(x)(p(x) - q(x)) \right| \leq \sum_{x \in \mathcal{X}} |f(x)| \cdot |p(x) - q(x)| \tag{10}$$

$$\leq \max_{x \in \mathcal{X}} |f(x)| \cdot \sum_{x \in \mathcal{X}} |p(x) - q(x)| \tag{11}$$

$$= 2 \cdot \max_{x \in \mathcal{X}} |f(x)| \cdot \|p - q\|_{\text{TV}}. \tag{12}$$

$\square$

## A.2 Main Theoretical Result

**Theorem A.2** (Diversity Preservation in DiCoFlex). *Let $p_\theta(x'|x, y', p, m)$ be the learned conditional normalizing flow and $\hat{q}(x'|x, y', d_{p,m})$ be the empirical distribution defined in Equation (6). Under assumptions (A1)-(A3) stated below, for $n$ samples $\{x_i'\}_{i=1}^n$ drawn from $p_\theta$, the expected diversity satisfies:*

$$\mathbb{E}_{p_\theta}[D(\{x_i'\}_{i=1}^n)] \geq \sigma_{\min}(1 - K^{1-n}) - \Delta\sqrt{2\epsilon}, \tag{13}$$

*where $D(\{x_i'\}_{i=1}^n) = \min_{i \neq j} \|x_i' - x_j'\|$ denotes the minimum pairwise distance among the generated counterfactuals.*

### Assumptions

**(A1)** *Minimum neighborhood separation*: The neighborhood $N(x, y', d_{p,m}, K)$ contains $K$ distinct points with minimum pairwise distance $\sigma_{\min} > 0$. Specifically, for any $x_i, x_j \in N(x, y', d_{p,m}, K)$ with $i \neq j$,

$$\|x_i - x_j\| \geq \sigma_{\min}. \tag{14}$$

**(A2)** *Training convergence*: The normalizing flow is trained sufficiently close to the empirical distribution, satisfying:

$$D_{\text{KL}}(\hat{q}(x'|x, y', d_{p,m}) \| p_\theta(x'|x, y', p, m)) \leq \epsilon \tag{15}$$

for all $(x, y', p, m)$ in the training distribution.

**(A3)** *Bounded support*: The support of $p_\theta(\cdot \mid x, y', p, m)$ lies in a set of finite diameter $\Delta$, i.e., for any $x', x''$ in the support,

$$\|x' - x''\| \leq \Delta. \tag{16}$$

This implies that $0 \leq D(\{x_i'\}_{i=1}^n) = \min_{i \neq j} \|x_i' - x_j'\| \leq \Delta$.

*Proof.* **Step 1: Lower bound on empirical diversity.** Consider the empirical distribution $\hat{q}(x'|x, y', d_{p,m})$ which samples uniformly from $K$ distinct neighbors. When sampling $n$ points with replacement from these $K$ points, the probability of selecting at least two distinct points is:

$$P_n(K) = 1 - K \cdot \left( \frac{1}{K} \right)^n = 1 - K^{1-n}. \tag{17}$$

Given assumption (A1), whenever two distinct points are selected, their distance is at least $\sigma_{\min}$. Therefore:

$$\mathbb{E}_{\hat{q}}[D(\{x'_i\}_{i=1}^n)] \geq \sigma_{\min} \cdot P_n(K) = \sigma_{\min}(1 - K^{1-n}). \tag{18}$$

**Step 2: Relating KL divergence to total variation distance.** By Pinsker's inequality [6, 40], which states that $\|p - q\|_{\mathrm{TV}} \leq \sqrt{\frac{1}{2} D_{\mathrm{KL}}(q \| p)}$ for any two probability distributions $p$ and $q$, the total variation distance is bounded by the KL divergence:

$$\|p_\theta(\cdot | x, y', p, m) - \hat{q}(\cdot | x, y', d_{p,m})\|_{\mathrm{TV}} \leq \sqrt{\frac{1}{2} D_{\mathrm{KL}}(\hat{q} \| p_\theta)} \leq \sqrt{\frac{\epsilon}{2}}. \tag{19}$$

**Step 3: Bounding the difference in expected diversity.** The diversity measure $D(\{x'_i\}_{i=1}^n)$ is bounded above by $\Delta$ (assumption A3). Applying Lemma A.1:

$$|\mathbb{E}_{p_\theta}[D] - \mathbb{E}_{\hat{q}}[D]| \leq 2 \cdot \Delta \cdot \|p_\theta - \hat{q}\|_{\mathrm{TV}} \leq 2\Delta \sqrt{\frac{\epsilon}{2}} = \Delta \sqrt{2\epsilon}. \tag{20}$$

**Step 4: Deriving the lower bound.** Considering the worst-case scenario where $p_\theta$ produces less diversity than $\hat{q}$:

$$\mathbb{E}_{p_\theta}[D(\{x'_i\}_{i=1}^n)] \geq \mathbb{E}_{\hat{q}}[D(\{x'_i\}_{i=1}^n)] - \Delta \sqrt{2\epsilon}. \tag{21}$$

Substituting the result from Step 1 yields the desired bound:

$$\mathbb{E}_{p_\theta}[D(\{x'_i\}_{i=1}^n)] \geq \sigma_{\min}(1 - K^{1-n}) - \Delta \sqrt{2\epsilon}. \tag{22}$$

$\square$

## A.3 Implications

**Diversity Guarantee** Theorem A.2 establishes that DiCoFlex maintains strong diversity guarantees. For sufficiently large $K$, the probability term $(1 - K^{1-n})$ approaches 1, ensuring that the expected diversity is approximately $\sigma_{\min} - \Delta \sqrt{2\epsilon}$. The theorem shows that the learned distribution $p_\theta$ recovers nearly the full empirical diversity, with deviation bounded by the training error $\epsilon$.

**Impact of Training Quality** As training progresses and $\epsilon \to 0$, the perturbation term $\Delta \sqrt{2\epsilon}$ vanishes, and the diversity of counterfactuals generated by the learned flow converges to that of the empirical distribution. This theoretical result validates the practical observation in Section 4 that DiCoFlex achieves superior diversity metrics compared to baseline methods.

**Sample Efficiency** The bound demonstrates that generating $n$ samples from DiCoFlex provides diversity guarantees that scale favorably with the number of neighbors $K$ used during training. For fixed $n$ and increasing $K$, the probability of obtaining diverse samples $(1 - K^{1-n})$ approaches 1 exponentially fast, explaining the effectiveness of our KNN-based training approach.

## A.4 Validity Guarantees

Our method provides inherent validity guarantees by construction. Since $\hat{q}(x' | x, y', d)$ (Equation (6)) exclusively samples from the set of $K$-nearest neighbors $N(x, y', d, K)$, where $h(x_i) = y'$, we have the following deterministic property:

**Proposition A.3** (Validity)**.** *For all training samples drawn from $\hat{q}(x' | x, y', d)$, the classification model returns the target class with probability one:*

$$p(y' | x') = 1, \quad \forall x' \sim \hat{q}(x' | x, y', d), \tag{23}$$

*where $p(y' | x')$ is the class probability returned by the classification model $h(\cdot)$.*

*Proof.* By definition of $\hat{q}$ in Equation (6), we have $\hat{q}(x' | x, y', d) > 0$ only if $x' \in N(x, y', d, K)$. The neighborhood $N(x, y', d, K)$ consists exclusively of training examples for which $h(x') = y'$ by construction. Therefore, $p(y' | x') = 1$ for all samples from $\hat{q}$. $\square$

This theoretical property explains why DiCoFlex achieves high validity in all datasets in Table 1 while being orders of magnitude faster than optimization-based methods. The normalizing flow trained on these valid samples inherits this property in expectation:

$$\mathbb{E}_{x' \sim p_\theta}[p(y'|x')] \approx 1, \tag{24}$$

with the approximation quality depending on the training convergence (assumption A2 in Theorem A.2).

### A.5 Proximity Guarantees

Our KNN sampling mechanism provides an implicit theoretical guarantee on proximity without requiring explicit proximity optimization.

**Proposition A.4** (Proximity Upper Bound). *For any counterfactual $x'$ sampled from $\hat{q}(x'|x, y', d)$ (Equation* (6)*), we have:*

$$d(x, x') \leq d(x, x_K^{y'}), \tag{25}$$

*where $x_K^{y'}$ is the $K$-th nearest neighbor of $x$ in class $y'$ under distance measure $d(\cdot, \cdot)$.*

*Proof.* By definition of $\hat{q}$ in Equation (6), we have $\hat{q}(x'|x, y', d) > 0$ if and only if $x' \in N(x, y', d, K)$. The set $N(x, y', d, K)$ contains exactly the $K$ nearest neighbors of $x$ in class $y'$ according to distance $d$. Therefore, any $x'$ with non-zero probability under $\hat{q}$ must satisfy:

$$d(x, x') \leq \max_{x_i \in N(x, y', d, K)} d(x, x_i) = d(x, x_K^{y'}), \tag{26}$$

as otherwise $x'$ would not be among the $K$ nearest neighbors and would have $\hat{q}(x'|x, y', d) = 0$. $\quad\square$

This provides an upper bound on proximity without requiring iterative optimization. The learned flow $p_\theta$ approximately respects this bound, with deviations controlled by the training error $\epsilon$ (assumption A2). The flexibility to adjust the distance measure $d_{p,m}$ (Equation (8)) allows users to control proximity characteristics at inference time through the sparsity parameter $p$ and mask $m$.

### A.6 Plausibility Guarantees

We now establish theoretical guarantees for plausibility, addressing a key limitation of optimization-based methods that may generate out-of-distribution counterfactuals.

**Proposition A.5** (In-Distribution Guarantee). *Since the training distribution $\hat{q}(x'|x, y', d)$ only assigns non-zero probability to $K$-nearest neighbors from the training set $\mathcal{D}$ (Equation* (6)*), any sampled counterfactual $x'$ satisfies:*

$$p_{data}(x') \geq \min_{x_i \in \mathcal{D}} p_{data}(x_i) > 0, \tag{27}$$

*where $p_{data}$ denotes the true data distribution.*

*Proof.* By construction, $\hat{q}(x'|x, y', d)$ assigns uniform probability $\frac{1}{K}$ to each of the $K$ neighbors in $N(x, y', d, K) \subset \mathcal{D}$ and zero probability elsewhere. Therefore, any sample $x' \sim \hat{q}$ is an actual training point: $x' \in \mathcal{D}$. Since all training points are drawn from $p_{data}$, we have $p_{data}(x') > 0$ for all such $x'$. $\quad\square$

**Enhanced Plausibility with Sparse Constraints**  When using small $p$ values (e.g., $p = 0.01$) in the $L_p$ norm distance (Equation (7)), we restrict the neighborhood to samples with minimal feature changes. These neighbors typically lie in higher-density regions of the data manifold, as points with few modified features are more likely to remain within dense areas of the feature space. This explains the improved Local Outlier Factor (LOF) scores for DiCoFlex (p=0.01) compared to DiCoFlex (p=2.0) observed in Table 1.

The normalizing flow learns to interpolate between these guaranteed in-distribution points while preserving the manifold structure through its invertible transformations. This provides a principled approach to ensuring plausibility, as the flow is constrained to learn smooth deformations between verified in-distribution samples rather than arbitrary transformations that might venture into low-density regions.

# B Dataset Descriptions

This appendix provides detailed descriptions of the datasets used in our experimental evaluation. Table 4 summarizes the key characteristics of each dataset, including the number of samples used for our experiments, number of numerical and categorical features, and number of classes.

Table 4: Dataset characteristics and statistics.

| Dataset | Samples | # Feat | # Num | # Cat | Target Class Dist. |
|---------|---------|--------|-------|-------|--------------------|
| Lending Club | 30,000 | 12 | 8 | 4 | 21.8% default |
| Give Me Some Credit | 30,000 | 9 | 6 | 3 | 6.7% yes |
| Bank Marketing | 30,000 | 16 | 7 | 9 | 11.3% yes |
| Credit Default | 27,000 | 23 | 14 | 9 | 22.1% yes |
| Adult Census | 32,000 | 12 | 4 | 8 | 24.5% ≥50K |

## B.1 Lending Club

The Lending Club dataset [15] contains detailed information about loans issued through the Lending Club peer-to-peer lending platform. It includes borrower characteristics (such as credit score, annual income, employment length), loan specifics (loan amount, interest rate, purpose), and performance indicators (payment status, delinquency). The binary classification task is to predict whether a loan will be fully paid or charged off (default). This dataset is particularly relevant for financial counterfactual explanations as it represents real-world credit risk assessment scenarios where understanding model decisions is crucial for both borrowers and lenders.

## B.2 Give Me Some Credit

The Give Me Some Credit dataset [28] contains anonymized records of credit users with features such as debt-to-income ratio, number of times delinquent, monthly income, age, and number of open credit lines. The target variable indicates whether a user experienced a serious delinquency (more than 90 days overdue) within the previous two years. This dataset provides insight into credit risk prediction in consumer finance, where counterfactual explanations can offer actionable guidance to consumers looking to improve their creditworthiness.

## B.3 Bank Marketing

The Bank Marketing dataset [21] contains information from a direct marketing campaign conducted by a Portuguese banking institution. The features include client data (age, job, marital status, education), campaign contact information (communication type, day, month), economic indicators, and previous campaign outcomes. The prediction task is to determine whether a client will subscribe to a term deposit. This dataset represents a real-world marketing scenario where understanding model decisions can improve campaign efficiency and provide insights for personalized marketing strategies.

## B.4 Credit Default

The Credit Default dataset [47] contains information on credit card clients in Taiwan, including demographic factors, credit data, payment history, and bill statements. The target variable indicates whether the client defaulted on their payment in the following month. With 23 features (14 numerical and 9 categorical), this dataset presents complex feature interdependencies common in financial data. The dataset is valuable for counterfactual explanation research because it represents real-world credit risk assessment with diverse feature types and nonlinear relationships.

## B.5 Adult

The Adult Census dataset [3] contains demographic information extracted from the 1994 U.S. Census database. Features include age, education, occupation, work hours per week, and capital gain/loss. The binary classification task is to predict whether an individual's income exceeds $50,000 per year.

This dataset is widely used in fairness and explainability research, as it contains sensitive attributes like race, gender, and age, making it valuable for studying how counterfactual explanations handle demographic factors.

## C   Evaluation Metrics

We evaluate counterfactual quality using metrics that correspond to key desiderata. For a test set $\mathcal{X}_{\text{test}}^0 = \{\mathbf{x}_n^0 | h(\mathbf{x}_n^0) = 0\}_{n=1}^N$, we generate set of $N$ counterfactuals $\mathcal{X}'^1 = \{\mathbf{x}_n'^1\}_{n=1}^N$ for class 1 with the evaluated model.

**Validity** measures the success rate of changing model predictions:

$$\text{Validity } (\uparrow) = \frac{N_{\text{val}}}{N} = \frac{1}{N} \sum_{n=1}^N \mathbb{I}(h(\mathbf{x}_n') = 1) \tag{28}$$

**Sparsity** quantifies feature modifications, separately for categorical and numerical features:

$$\text{Sparsity Cat/Num } (\downarrow) = \frac{1}{N_{\text{val}}} \sum_{n=1}^{N_{\text{val}}} \frac{\|\mathbf{x}_{n,\text{cat/num}}^0 - \mathbf{x}_{n,\text{cat/num}}'^1\|_0}{D_{\text{cat/num}}}, \tag{29}$$

where $\mathbf{x}_{n,\text{cat/num}}^0$ represents $n$-th generated counterfactual reduced to categorical numerical attributes.

For continuous features, we use $\epsilon$**-sparsity**, counting features where the relative change from the original value exceeds threshold $\epsilon = 0.05$:

$$\epsilon\text{-Sparsity cont. } (\downarrow) = \frac{1}{N_{\text{val}}} \sum_{n=1}^{N_{\text{val}}} \frac{1}{|D_{\text{cont}}|} \sum_{d \in D_{\text{cont}}} \mathbb{I}\left( \frac{|x_{n,d} - x_{n,d}'|}{|x_{n,d}|} > \epsilon \right) \tag{30}$$

where $x$ denotes original values and $x'$ is the counterfactual.

**Proximity** measures distance to original instances using Manhattan for continuous features:

$$\text{Proximity Num } (\downarrow) = \frac{1}{N_{\text{val}}} \sum_{n=1}^{N_{\text{val}}} \|\mathbf{x}_{n,\text{num}}^0 - \mathbf{x}_{n,\text{num}}'^1\|_1 \tag{31}$$

**Predictive Performance** is assessed via **Validity** and **Classif. prob.** (model confidence in target prediction).

**Diversity** is measured by **Hypervol. log scale** [24], which quantifies spread and coverage in objective space.

**Plausibility** is evaluated with **LOF log scale** (Local Outlier Factor from training distribution) and **Log Density** (probability under training data distribution).

Lower values are better for sparsity, proximity, and plausibility metrics ($\downarrow$), while higher values are better for validity, model confidence, and diversity ($\uparrow$).

## D   Generative Model Selection

We conducted an ablation study to select the most suitable normalizing flow architecture for our framework, comparing different models based on negative log-likelihood (NLL) on the Adult dataset.

Table 5: Negative log-likelihood comparison of generative models on the Adult dataset. Lower values indicate better performance.

|  | MAF | NICE | RealNVP | KDE |
|---|---|---|---|---|
| NLL | **-43.2998** | 26.6644 | 26.5827 | 30.5120 |

We compared three normalizing flow architectures: MAF [26], NICE [8], and Real NVP [9], as well as KDE as a non-parametric baseline. MAF significantly outperformed all alternatives, while NICE and RealNVP showed comparable performance, and KDE exhibited the poorest results.

MAF's superior performance stems from its autoregressive structure, which enables more expressive transformations by conditioning each dimension on previously transformed dimensions. This property is crucial for accurately modeling the conditional distribution of counterfactual explanations. Based on these results, we selected MAF as the architecture for DiCoFlex, contributing significantly to its ability to generate high-quality, diverse counterfactual explanations.

# E    Details of training algorithm

---

**Algorithm 1** Training procedure

---

**Require:** number of steps $T$, training examples $\mathcal{X}$, classification model $h(\cdot)$, prior class distribution $\boldsymbol{\pi}$, set of sparsity levels $\mathcal{P}$, set of considered masking $\mathcal{M}$, number of nearest neighbors $K$
Initialize $\theta_0$
**for** $t = 1$ to $T$ **do**
    Sample $\mathbf{x} \sim \mathcal{X}$ and class $y' \sim \boldsymbol{\pi} \setminus \{h(\mathbf{x})\}$
    Sample sparsity level $p \sim \mathcal{P}$ and mask $m \sim \mathcal{M}$;
    Sample $K$ counterfactuals $\mathbf{x}' \sim \hat{q}(\mathbf{x}'|\mathbf{x}, y', d_{p,\mathbf{m}})$ given by (6);
    Update parameters $\theta_t$ by optimizing $\mathcal{Q}$ given by (4);
**end for**
**return** $x_0$

---

The training procedure is outlined in Algorithm 1. In each training iteration, a data example $\mathbf{x}$ is drawn from the dataset $\mathcal{X}$, and a target class $y'$ is sampled from the class distribution excluding the current class of $\mathbf{x}$, i.e., $\boldsymbol{\pi} \setminus h(\mathbf{x})$. Subsequently, a sparsity level $p$ is selected from a predefined set $\mathcal{P}$, along with a corresponding masking vector $\mathbf{m}$ from the set $\mathcal{M}$.

Next, the set of $K$ nearest neighbors, denoted by $N(\mathbf{x}, y', d_{p,\mathbf{m}}, K)$, is computed using the distance measure $d_{p,\mathbf{m}}$ defined in (8). A counterfactual example $\mathbf{x}'$ is then sampled from the distribution $\hat{q}(\mathbf{x}' \mid \mathbf{x}, y', d_{p,\mathbf{m}})$ as defined in (6). Finally, the parameters $\theta_t$ are updated via a gradient-based optimization procedure aimed at maximizing the objective in (4), with conditioning on both $p$ and $\mathbf{m}$.

## E.1    Training Details and Hyperparameter Exploration

We report additional details of the DiCoFlex training process and hyperparameter configuration used in all experiments. All experiments were conducted on a GPU workstation equipped with an NVIDIA RTX 4090 (24 GB VRAM) and an AMD Ryzen Threadripper PRO 5975WX CPU with 256 GB RAM. Each model was trained for a maximum of 1000 epochs using the Adam optimizer ($\mathrm{lr} = 10^{-4}$) and early stopping with a patience of 300 epochs based on the validation objective.

**Training times.**    Training times for all benchmark datasets are visualized in Figure 3. Each subfigure corresponds to one dataset (Adult, Bank Marketing, Default, Give Me Some Credit, and Lending Club) and reports the wall-clock time required to complete full training, including early stopping. The training time primarily depends on the dataset size and feature dimensionality.

**Hyperparameter exploration.**    We explored several key hyperparameters influencing model behavior and training stability:

Number of nearest neighbors $K \in \{8, 16, 32\}$

Actionability penalty $\alpha \in \{1, 10, 1000\}$

MAF hidden features $\in \{16, 32, 64\}$

MAF hidden layers $\in \{2, 5\}$

For each combination, models were trained under identical conditions with early stopping. Hyperparameter selection was guided by validation performance averaged across validity, proximity, and

sparsity metrics, favoring configurations that balanced these criteria without overfitting. Unless otherwise stated, the final configuration used $K=16$, $\alpha=10$, 32 hidden features, and 5 hidden layers.

**Sensitivity analysis.** To verify robustness, we performed a limited sensitivity analysis on the *Adult* dataset, varying each hyperparameter within the ranges listed above. Results confirmed that DiCoFlex remains stable across a wide range of $K$, $\alpha$, and model capacities, with minimal variation in validity and plausibility metrics. Detailed quantitative sensitivity results are provided in Appendix E.2.

# F  Limitations of our method

DiCoFlex is deliberately designed for tabular data with mixed feature types, the most common domain for counterfactual explanations in high-stakes decision-making contexts. Although this focus enables strong performance where interpretability is most needed, extending to other data modalities would require architectural adaptations. Our approach exhibits natural trade-offs between competing explanation criteria, favoring diversity and plausibility while maintaining competitive performance in sparsity metrics. The computational efficiency during inference requires an initial training investment, though this one-time cost enables subsequent real-time applications that outperform existing methods by orders of magnitude.

Recent work has demonstrated that counterfactual explanations can be exploited for model extraction attacks, as they reveal decision boundary information [1, 45]. Furthermore, diverse counterfactual explanations can particularly enhance extraction effectiveness. However, DiCoFlex's design may offer partial protection. Unlike methods exploring arbitrary feature space regions, the learned normalizing flow captures only the conditional distribution over plausible, proximal counterfactuals, potentially limiting information available for extraction. Nevertheless, some extraction risk remains. DiCoFlex's diversity may still allow adversaries to triangulate decision boundaries from multiple sparse directions.

DiCoFlex generates counterfactuals based on proximity and plausibility without explicit fairness constraints. If the underlying model exhibits demographic biases, DiCoFlex may generate counterfactuals that reflect or amplify these biases. Future work should investigate incorporating fairness-aware constraints into the normalizing flow training to ensure protected attributes are handled appropriately.

As DiCoFlex trains on $K$-nearest neighbors from labeled training data (Equation 6), counterfactual quality depends on label accuracy.

# G  Context Vector Architecture

The conditional normalizing flow $p_\theta(\mathbf{x}'|\mathbf{x}, y', p, \mathbf{m})$ requires a mechanism to incorporate conditioning information during the generative process. We achieve this through a context vector that concatenates all conditioning inputs:

$$\mathbf{c} = [\mathbf{x}, y', p, \mathbf{m}] \in \mathbb{R}^{d+1+1+d}, \tag{32}$$

where $\mathbf{x} \in \mathbb{R}^d$ is the original instance, $y' \in \{0, 1\}$ is the target class (one-hot encoded for multi-class scenarios), $p \in \mathbb{R}$ is the sparsity parameter controlling the $L_p$ norm, and $\mathbf{m} \in \{0, 1\}^d$ is the binary actionability mask indicating which features should remain unchanged.

Following the Masked Autoregressive Flow (MAF) architecture [26], this context vector $\mathbf{c}$ is passed to each layer of the flow through the affine coupling transformations. Specifically, in MAF, each transformation layer computes:

$$\mathbf{z}_i = \mathbf{x}'_i \odot \exp(\alpha_i(\mathbf{x}'_{<i}, \mathbf{c})) + \mu_i(\mathbf{x}'_{<i}, \mathbf{c}), \tag{33}$$

where $\alpha_i$ and $\mu_i$ are neural networks that produce scale and shift parameters conditioned on both previous dimensions $\mathbf{x}'_{<i}$ and the context vector $\mathbf{c}$. This autoregressive conditioning allows the flow to learn expressive transformations that respect the user-specified constraints encoded in $\mathbf{c}$.

The key advantage of this architecture is that it enables flexible control at inference time: users can adjust $p$ and $\mathbf{m}$ dynamically to generate counterfactuals satisfying different sparsity and actionability

requirements without retraining the model. The flow learns to map these constraint specifications to appropriate regions of the counterfactual distribution during training, as described in Algorithm 1.

## H    Computational Resources

Our experimental framework utilized Python [42] as the primary programming language, Additionally, the open-source machine learning library PyTorch [27] is used to implement DiCoFlex. All experiments were conducted on a GPU cluster equipped with a GeForce RTX 4090 graphics card (24 GB VRAM) and an AMD Ryzen Threadripper PRO 5975WX 32-core processor, with 256 GB of available RAM. These resources provide sufficient computational power and processing speed to meet the requirements of our algorithm.

## I    Additional experimental results

### I.1    Runtime Comparison Across Datasets

Figure 3 displays runtime comparisons between DiCoFlex and baseline methods across all datasets, with time shown on a logarithmic scale. Both variants of DiCoFlex consistently achieve substantially faster execution times for counterfactual generation, while competing approaches demand processing times that are orders of magnitude longer. This substantial performance advantage stems from fundamental architectural differences. DiCE [22] performs separate optimization procedures for each explanation. CCHVAE [28] requires expensive latent space searching. Similarly, Wachter [44], ReViSE [16] and TABCF [25] rely on iterative or gradient-based optimization procedures that scale poorly with the number of counterfactuals generated. In contrast, DiCoFlex leverages conditional normalizing flows trained solely on labeled data to generate multiple diverse counterfactuals in a single forward pass. By eliminating iterative optimization procedures and model access at inference time, DiCoFlex enables real-time counterfactual generation.

### I.2    Impact of Actionability Constraints

Table 6 presents the evaluation of counterfactual explanations generated by DiCoFlex with different actionability constraints imposed through feature masks. The analysis reveals complex relationships between masking constraints and evaluation metrics. In particular, the application of different masks results in varying impacts across metrics without following a consistent directional pattern.

Mask 1, which prevents modifications to Capital Gain and Capital Loss, achieves the lowest continuous proximity score but exhibits reduced diversity, as indicated by its hypervolume score. In contrast, mask 4 (restricting Sex and Native Country modifications) yields the highest hypervolume while maintaining moderate performance across other metrics. Mask 3 (constraining Age) demonstrates high classification probability but the highest proximity score, indicating a greater deviation from the original instances.

The observed results indicate that actionability constraints introduce complex trade-offs that do not follow simple patterns. The absence of consistent correlations between metrics under different masking configurations suggests that performance characteristics are highly dependent on the specific constraints applied rather than adhering to predictable trade-off relationships.

These findings further validate DiCoFlex flexibility in selecting various user-defined constraints without retraining, while demonstrating that the selection of appropriate constraints should be guided by domain-specific requirements rather than general optimization principles. The mechanism enables the practical customization of counterfactual explanations according to specific application needs, where different feature restrictions may be necessary due to legal, ethical, or practical considerations.

### I.3    Statistical Uncertainty of Results

To assess the robustness of the reported metrics, we compute standard deviations across counterfactual sets generated per factual instance. For each test point, we produce multiple counterfactuals and evaluate the per-instance metric variance; the reported values represent the mean standard deviation across the test set. Note that the hypervolume metric operates at the set level and thus does not admit a standard deviation estimate.

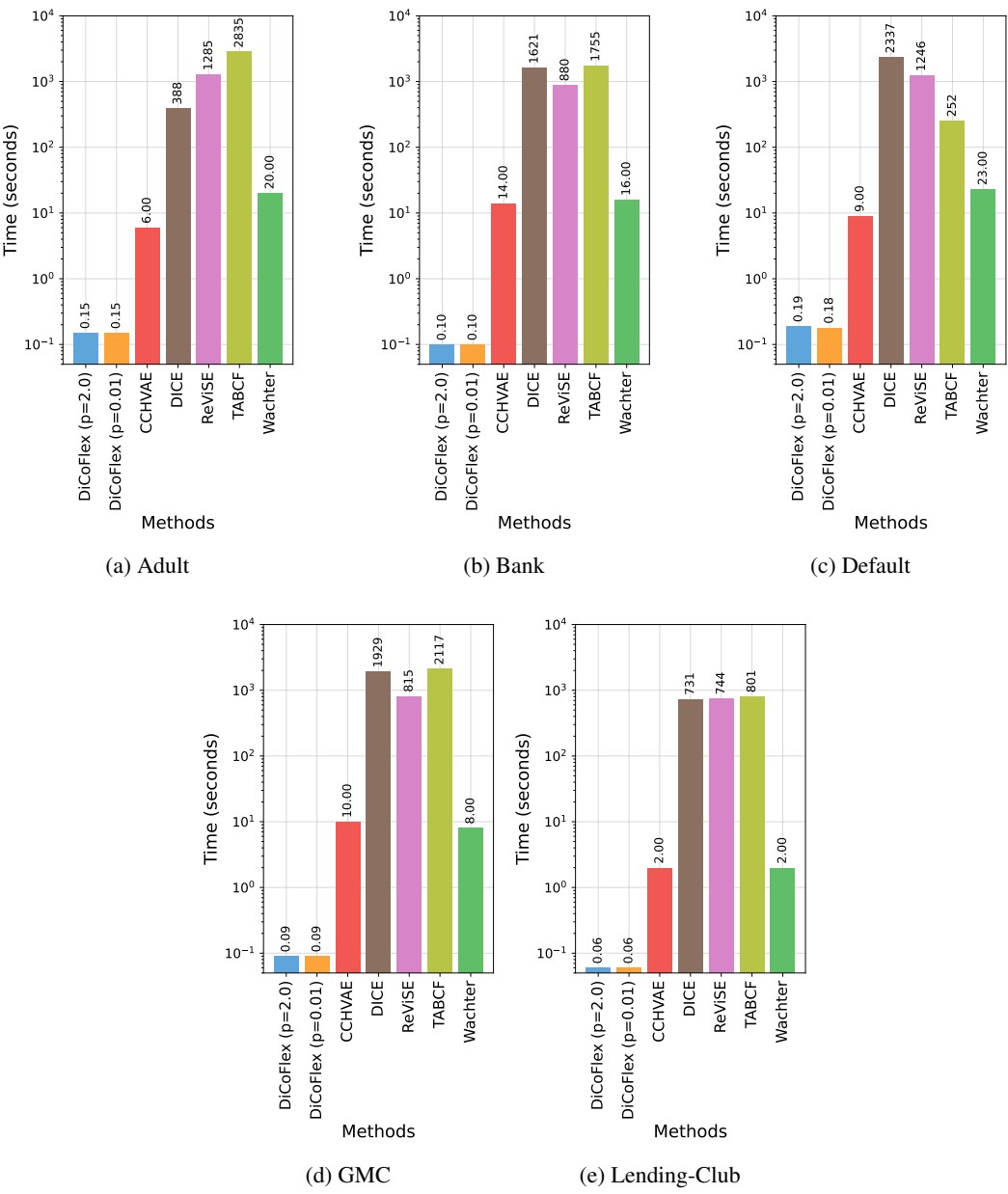

Figure 3: Visualization of runtime of DiCoFlex method and other baseline methods.

*Notation:* Standard deviations are denoted directly as metric values in this table (no $\pm$ notation used for compactness). Metrics are averaged across counterfactuals for each instance and then aggregated across the test set.

## I.4  German Credit Dataset Results

To evaluate the generalization of DiCoFlex to smaller tabular datasets, we conduct an additional experiment on the *German Credit* dataset (1,000 samples, 20 features, binary target). Despite the limited data availability, DiCoFlex maintains consistent performance across all evaluation metrics, demonstrating robustness in low-data regimes.

Even under constrained data conditions, both DiCoFlex variants (DiCoFlex (p=2.0) and DiCoFlex (p=0.01)) achieve perfect validity and strong plausibility (lowest LOF), confirming their ability to learn consistent counterfactual manifolds from limited examples. The results also show that the

Table 6: Influence of imposing actionability constraints on the metrics used to evaluate counterfactual explanations.

| Model | Classif. prob. ↑ | Proximity cont. ↓ | Sparsity cat. ↓ | $\epsilon$-sparsity cont. ↓ | LOF log scale ↓ | Hypervol. log scale ↑ |
|---|---|---|---|---|---|---|
| mask 1 | 0.987 | **0.496** | 0.557 | **0.483** | **1.881** | 0.881 |
| mask 2 | 0.980 | 0.553 | 0.555 | 0.517 | 2.364 | 1.703 |
| mask 3 | **0.993** | 0.627 | 0.568 | 0.513 | 2.310 | 2.411 |
| mask 4 | 0.992 | 0.517 | 0.590 | 0.520 | 2.327 | **2.640** |
| unconstrained | 0.998 | 0.581 | **0.515** | 0.498 | 2.156 | 2.036 |

Table 7: Standard deviations of evaluation metrics across datasets and methods. Values indicate variability across generated counterfactuals for each instance. Lower values correspond to more stable generation behavior. Standard deviations are shown for all metrics except *Hypervolume*, which is computed at the set level and thus not associated with instance-wise variance.

| Dataset | Model | Validity ↑ | Classif. prob. ↑ | Proximity cont. ↓ | Sparsity cat. ↓ | $\epsilon$-sparsity cont. ↓ | LOF log scale ↓ |
|---|---|---|---|---|---|---|---|
| Lending Club | DiCoFlex (p=2.0) | 0 | **0.012** | 0.439 | 0.198 | 0.087 | 0.076 |
| | DiCoFlex (p=0.01) | 0 | 0.031 | **0.432** | 0.209 | 0.100 | **0.053** |
| | CCHVAE | 0 | 0.368 | **0.401** | 0.049 | 0.066 | 0.119 |
| | DiCE | 0 | 0.037 | 1.358 | **0.004** | **0.053** | 0.167 |
| Adult | DiCoFlex (p=2.0) | 0 | 0.011 | 0.518 | 0.099 | 0.127 | **0.256** |
| | DiCoFlex (p=0.01) | 0 | **0.000** | **0.352** | 0.106 | **0.096** | 0.812 |
| | CCHVAE | 0 | 0.102 | 0.646 | 0.049 | 0.189 | 0.976 |
| | DiCE | 0 | **0.000** | 1.464 | **0.017** | 0.264 | 0.915 |
| Bank | DiCoFlex (p=2.0) | 0 | 0.025 | 0.571 | 0.100 | 0.111 | 0.016 |
| | DiCoFlex (p=0.01) | 0 | 0.119 | **0.466** | 0.088 | **0.073** | **0.011** |
| | CCHVAE | 0 | 0.053 | 0.606 | 0.061 | 0.090 | 0.078 |
| | DiCE | 0 | **0.012** | 1.162 | **0.056** | 0.082 | 0.139 |
| Default | DiCoFlex (p=2.0) | 0 | 0.021 | 0.346 | 0.138 | **0.071** | 0.038 |
| | DiCoFlex (p=0.01) | 0 | **0.012** | **0.324** | 0.171 | 0.085 | **0.034** |
| | CCHVAE | 0 | 0.043 | 0.273 | **0.097** | 0.076 | 0.047 |
| | DiCE | 0 | 0.097 | 1.203 | 0.141 | 0.113 | 0.073 |
| GMC | DiCoFlex (p=2.0) | 0 | **0.032** | 0.535 | 0.235 | **0.058** | 0.061 |
| | DiCoFlex (p=0.01) | 0 | 0.037 | 0.694 | 0.229 | 0.059 | 0.099 |
| | CCHVAE | 0 | 0.052 | **0.507** | 0.148 | 0.076 | 0.064 |
| | DiCE | 0 | 0.072 | 1.021 | **0.048** | 0.073 | **0.019** |

Table 8: Comparison of counterfactual generation methods on the *German Credit* dataset. Lower proximity, sparsity, $\epsilon$-sparsity, and LOF indicate better results; higher validity, classification probability, and hypervolume indicate improved performance.

| Dataset | Model | Validity ↑ | Classif. prob. ↑ | Proximity cont. ↓ | Sparsity cat. ↓ | $\epsilon$-sparsity cont. ↓ | LOF log ↓ | Hypervol. log ↑ |
|---|---|---|---|---|---|---|---|---|
| German Credit | DiCoFlex (p=2.0) | **1.000** | **0.833** | 0.639 | 0.553 | 0.756 | 0.0465 | **-6.030** |
| | DiCoFlex (p=0.01) | **1.000** | **0.833** | 0.647 | 0.556 | **0.750** | 0.0443 | -6.115 |
| | CCHVAE | **1.000** | 0.650 | **0.610** | 0.538 | 0.825 | 0.0747 | -8.878 |
| | DiCE | 0.954 | 0.808 | 1.143 | **0.257** | 0.756 | 0.1936 | -7.456 |
| | ReViSE | 0.673 | 0.731 | 0.841 | 0.523 | 0.742 | 0.1731 | – |
| | TABCF | 1.000 | 0.912 | 0.829 | 0.348 | 0.559 | 0.0508 | – |
| | Wachter | 0.356 | 0.508 | 1.633 | 0.000 | 0.952 | 0.2529 | – |

hypervolume metric remains competitive, suggesting that diversity is preserved even when the training data distribution is relatively sparse.

These findings highlight that DiCoFlex generalizes effectively to smaller datasets, further reinforcing its practicality for real-world applications where large annotated samples may be unavailable.

