# OpenReview forum: "DiCoFlex: Model-Agnostic Diverse Counterfactuals with Flexible Control"
_NeurIPS.cc/2025/Conference — NeurIPS 2025 poster_

### Official Review · Reviewer_jqT4 · 2025-06-16

**Clarity:** 1
**Significance:** 3
**Originality:** 2
**Rating:** 4
**Confidence:** 4

**Summary:**

This paper proposes a new generative approach for providing diverse counterfactual explanations. A key idea of the proposed method, called DiCoFlex, is to train a normalizing flow model by taking validity, proximity, and plausibility into account. In addition, DiCoFlex can handle sparsity and actionability constraints using a customized proximity measure without requiring retraining. Through experiments, the authors demonstrated that DiCoFlex was significantly faster than existing baselines, while maintaining comparable or superior quality of generated counterfactuals.

**Questions:**

1. In the experiments, the authors used five datasets with about 30000 samples. Can the proposed method perform similarly for smaller benchmark datasets, such as FICO and German?
1. In Section 3.4, the authors state, "While sparsity is often implicitly achieved for categorical variables due to the applied optimization scheme, enforcing sparsity on continuous features remains challenging." and "Since categorical attributes are commonly encoded by one-hot vectors, the above scheme affects mostly continuous features." What do these sentences mean?

**Ethical Concerns:**

["NO or VERY MINOR ethics concerns only"]

**Final Justification:**

Thank you for taking the time to answer all of my questions and concerns, and especially thank you for reporting additional experimental results.
Since the rebuttal addressed some of my concerns (e.g., training details and small datasets), I have decided to improve my score.

**Limitations:**

While the authors discussed the limitations of the proposed method in Appendix, there seems to be no discussion on its potential negative societal impacts. For example, I think the diverse counterfactual explanation poses a risk of model extraction [3, 4].

[3] Aïvodji et al.: Model extraction from counterfactual explanations. arXiv, 2020.

[4] Wang et al.: DualCF: Efficient Model Extraction Attack from Counterfactual Explanations. FAccT, 2022.

**Paper Formatting Concerns:**

I could not find any formatting issues in this paper.

**Quality:**

2

**Strengths And Weaknesses:**

Strengths:
1. The motivation of the proposed method is reasonable. I think it is an interesting idea to model the desiderata of counterfactual explanation, including not only plausibility but also actionability, by the normalizing flow.
1. The authors demonstrated well that the proposed method was significantly faster than the baselines without compromising the quality of the generated counterfactuals.

Weaknesses:
1. The presentation of the proposed method in Section 3 could be more clarified and improved. While Section 3 primarily focuses on modeling $q$ and learning it as $p_\theta$, it remains unclear how to generate counterfactuals. If my understanding is correct, the proposed method generates a set of counterfactuals $x'$ by sampling from a trained normalizing flow model $p_\theta(x' \mid x, y', p, m)$. However, I could not find how the proposed method ensures the diversity of the generated counterfactuals. The score function defined in Equation (6) was confusing for me because it is not directly used to generate counterfactuals.
1. Although the proposed method heavily relies on the normalizing flow, this paper lacks details on the training of the normalizing flows in the experiments. For example, the average running times of training the normalizing flow models are not reported in this paper. In addition, I could not find how the authors determined the hyperparameters of the proposed method, such as $K$, $\alpha$, number of training steps $T$, and model complexity (e.g., hidden layer sizes). I am concerned that the choice of these parameters may affect the performance of the proposed method.
1. (minor) I am concerned about the use of the $L_p$-norm with $p \in (0, 1)$ (e.g., $p=0.01$), as it may lead to instability in gradient-based training. I wonder if the sparsity can be sufficiently encouraged by the general $L_1$-norm.
1. (minor) Although the key technique of the proposed method is the normalizing flow, this paper appears to overlook existing methods that also leverage the normalizing flow [1, 2].

[1] Wielopolski et al.: Probabilistically Plausible Counterfactual Explanations with Normalizing Flows. arXiv, 2024.

[2] Furman et al.: Unifying Perspectives: Plausible Counterfactual Explanations on Global, Group-wise, and Local Levels. arXiv, 2024.

---

> ### Author Rebuttal · Authors · 2025-07-30
>
> Thank you for your detailed review and constructive feedback. We also appreciate your recognition of our method's key strengths: the novel use of normalizing flows to model counterfactual desiderata (including plausibility and actionability) and our demonstration that the proposed method achieves significant computational efficiency without compromising counterfactual quality.
>
> We address your concerns below:
> # Presentation and Method Clarity
>
> We thank the reviewer for raising this important point. In DiCoFlex, counterfactuals are generated by sampling multiple noise vectors from the base distribution (assumed to be Gaussian) that are further processed using the inverse transformation of the normalizing flow to transform them to counterfactuals. Because we utilize the conditioned variant of the normalizing flow, this process is guided by the initial example $x$, the target class $y'$, and the parameters $p$ and $m$ that control sparsity and controlability. By modeling $p(x'|x,y,p,m')$ as a distribution rather than a point estimate, sampling multiple times naturally produces diverse counterfactuals. The normalizing flow learns the manifold of counterfactuals, and different samples explore different regions of this manifold. That diversity obtained in such a process was empirically confirmed in the evaluation using the Hypervolume metric in our experiments. The role of the score function given by Eq. (6) was just to show how the constraints representing the validity, proximity, and plausibility can be aggregated into one criterion (see more detailed explanation provided to reviewer TN7b). We will revise Section 3 to make the generation process and the role of the score function more explicit.
>
> Furthermore, we provide theoretical analysis of diversity guarantees for our method (See the answer for reviewer TN7b).
>
> # Training Details and Hyperparameters
> We conducted extensive hyperparameter exploration across the following ranges:
> - K: \[8, 16, 32\]
> - alpha: \[1, 10, 1000\]
> - MAF hidden features: \[16, 32, 64\]
> - MAF hidden layers: \[2, 5\]
> - Training configuration: 1000 epochs with early stopping patience of 300 epochs
>
> Training times are reported in a table below:
> | Dataset | Training Time (seconds) |
> |---|---|
> | Bank | 1644.73 |
> | Adult | 5340.32 |
> | GMC | 5537.64 |
> | LendingClub | 5453.87 |
> | Default | 3611.40 |
> | German Credit | 95.69 |
>
> The final hyperparameters were selected based on validation performance across multiple datasets, balancing counterfactual quality metrics (validity, proximity, plausibility) with computational efficiency. We will include a detailed hyperparameter sensitivity analysis in the final version, showing how these choices affect performance across different datasets and metrics.
>
> # Technical Concerns
> **$L_p$ Norm Stability**: Your concern about gradient instability with $p \in (0,1)$ is understandable but doesn't apply here. **No gradients pass through this operation.** The $L_p$ norm is used only during the k-NN selection phase to define the ground-truth counterfactuals for training. The sparsity control mechanism operates at the data preparation level, not during gradient-based optimization of the flow model.
>
> **Related Work**: Thank you for highlighting the relevant papers [1,2]. Our key distinction is modeling the conditional probability distribution $p(x'|x,y')$ where both original instance x and target class y' condition the generation. In contrast, the cited works use flows to model $p(x'|y')$ and assess plausibility. Furthermore those methods have limitations regarding processing categorical variables, therefore it is impossible to include them to our benchmark. We will add detailed comparisons to our related work section.
>
> # Experimental Questions
>
> ## Smaller Datasets
> We evaluated our method on the German Credit dataset (1,000 samples) as requested. The results demonstrate that our approach maintains excellent performance on smaller datasets:
>
> | Dataset | Model | Validity ↑ | Classif. prob. ↑ | Proximity cont. ↓ | Sparsity cat. ↓ | ε-sparsity cont. ↓ | LOF log scale ↓ | Hypervol. log scale ↑ |
> |---------|-------|------------|-------------------|-------------------|-----------------|-------------------|-----------------|----------------------|
> | German Credit | DiCoFlex | **1.000** | **0.833** | 0.639 | 0.553 | 0.756 | 0.0465 | **-6.030** |
> | | s-DiCoFlex | **1.000** | **0.833** | 0.647 | 0.556 | **0.750** | **0.0443** | -6.115 |
> | | CCHVAE | **1.000** | 0.650 | **0.610** | 0.538 | 0.825 | 0.0747 | -8.878 |
> | | DiCE | 0.954 | 0.808 | 1.143 | **0.257** | 0.756 | 0.1936 | -7.456 |
> | | ReViSE | 0.673 | 0.731 | 0.841 | 0.523 | 0.742 | 0.1731 | -- |
> | | TABCF | 1.000 | 0.912 | 0.829 | 0.348 | 0.559 | 0.0508 | -- |
> | | Wachter | 0.356 | 0.508 | 1.633 | 0.000 | 0.952 | 0.2529 | -- |
>
> Our method achieves perfect validity and maintains high classification probability while delivering the best hypervolume performance and competitive LOF scores. These results confirm that our approach scales effectively to smaller benchmark datasets, addressing the reviewer's concern about generalizability across dataset sizes.
>
> ## Sparsity Explanation
> Thank you for seeking clarification on these statements. We provide the following explanation. The statements in Section 3.4 refer to the natural behavior of our k-NN selection mechanism:
>
> **Why categorical sparsity is implicit:** The key insight lies in the distance metric behavior for different variable types. For categorical variables encoded as one-hot vectors, consider two binary vectors $x=(x_1,x_2)$ and $y=(y_1,y_2)$. The p-distance $d_p(x,y)$ equals 0 when $x_1=y_1$ and equals 2 otherwise, regardless of the chosen p value. This discrete jump property means that to transition between different categories (changing only one categorical variable), the optimization must traverse a fixed distance of 2, naturally limiting categorical changes.
>
> **Why continuous features need explicit control:** In contrast, continuous variables allow for gradual changes with smaller distances. The choice of p-norm becomes crucial here, as smaller p-values promote sparse changes by concentrating modifications on fewer coordinates.
> We acknowledge that alternative approaches, like dedicated preprocessing normalization, exist but are beyond the scope of this work.
>
> # Limitations and Societal Impact
> The reviewer raised an important point about model extraction risks [3,4]. We will expand our limitations section to address potential negative societal impacts, including risk of model extraction through diverse counterfactual explanations.
>
> We believe that these revisions will significantly improve the paper's clarity and completeness while addressing your technical concerns. Thank you for helping us to strengthen this work.

---

> > ### Comment · Reviewer_jqT4 · 2025-08-02
> >
> > Thank you for taking the time to answer all of my questions and concerns, and especially thank you for reporting additional experimental results.
> > Since the rebuttal addressed some of my concerns (e.g., training details and small datasets), I have decided to improve my score.

---

> > > ### Author Response · Authors · 2025-08-04
> > >
> > > Thank you for reconsidering our submission and improving your score after reviewing our rebuttal. We appreciate your constructive feedback and are pleased that our responses addressed your concerns regarding training details and evaluation on smaller datasets.
> > >
> > > We are committed to incorporating all requested improvements and clarifications into the camera-ready version. Your thorough review has enhanced the quality of our work and will make it more accessible to the research community.

---

### Official Review · Reviewer_CeoU · 2025-06-28

**Clarity:** 3
**Significance:** 3
**Originality:** 3
**Rating:** 5
**Confidence:** 4

**Summary:**

This paper introduces DiCoFlex, a conditional generative framework for model-agnostic counterfactual explanation. DiCoFlex is built upon conditional normalizing flows (specifically, Masked Autoregressive Flows) and trained using only labeled data, without requiring access to model gradients or internals. It enables generation of diverse, plausible, and constraint-aware counterfactuals in a single forward pass. The method is evaluated on five benchmark tabular datasets, where it outperforms or matches state-of-the-art baselines across key metrics including diversity, proximity, and plausibility.

**Questions:**

1. DiCoFlex vs. s-DiCoFlex: Since p is part of the conditioning context and can be set dynamically at inference time, what exactly distinguishes s-DiCoFlex from DiCoFlex beyond using different p values? Are they trained with different p ranges, or is this merely a naming convention?

2. Sensitivity analysis: Can the authors provide or discuss a broader sweep over p values to show how sparsity, plausibility, and proximity metrics vary as a function of p?

3. Clarify the wording around the sparsity effect of low p: the paper says “lower sparsity” is encouraged when p = 0.01, but the intended behavior is to encourage higher sparsity.

4. Context vector construction: Can the authors clarify how the conditioning vector (including p, mask, and target class) is structured and passed into the MAF layers?

**Ethical Concerns:**

["NO or VERY MINOR ethics concerns only"]

**Final Justification:**

The authors answered my questions and provided additional information on DiCoFlex vs. s-DiCoFlex. I am updating my score upwards, trusting the authors to better clarify these points in the final version of the paper.

**Limitations:**

Additional reflection on robustness to labeling noise or fairness impacts of constraint settings would further strengthen this part.

**Quality:**

3

**Strengths And Weaknesses:**

Strengths:

The approach is model-agnostic and does not require gradient access or retraining when user constraints change.

DiCoFlex supports dynamic inference-time control over constraints, which is both novel and practical.

The model is efficient: it generates multiple diverse counterfactuals per instance via a single forward pass.

Experimental results are extensive and well-structured across multiple datasets and evaluation metrics.

The paper is highly reproducible, with code, datasets, evaluation procedures, and a detailed training algorithm provided.



Weaknesses:

While DiCoFlex claims inference-time controllability of sparsity via parameter p, the comparison between DiCoFlex and s-DiCoFlex is somewhat confusing , the paper treats them as different variants even though they use the same model with different p values.

The paper does not provide a sensitivity analysis or sweep over different values of p, which could strengthen the claim of flexible constraint handling.

Statistical uncertainty (e.g., standard deviations, confidence intervals) is not reported in the evaluation.

The sparsity explanation in Section 4 contains unclear wording: it states “p = 0.01 encourages lower sparsity”, which contradicts the intended meaning (lower p encourages higher sparsity).

---

> ### Author Rebuttal · Authors · 2025-07-30
>
> Thank you for your thoughtful and constructive review. We appreciate your recognition of our method's strengths, particularly the model-agnostic approach, dynamic constraint control, and computational efficiency. We address your concerns below:
>
> # DiCoFlex vs. s-DiCoFlex Clarification
> You are correct that DiCoFlex and s-DiCoFlex represent the same model. s-DiCoFlex is simply DiCoFlex evaluated with p=0.01 during inference to demonstrate sparsity control. Users can dynamically adjust p at inference time. We acknowledge the separate naming is misleading and will revise the paper to clarify that these are configurations of a single model rather than distinct methods.
>
> # Sensitivity Analysis Over p Values
> We agree that a comprehensive sensitivity analysis strengthens our claims about flexible constraint handling. The table below presents results for different p values ($p \in \{{0.01, 0.08, 0.25, 1.0, 2.0}\}$) for the Adult dataset, demonstrating the smooth control our method provides over the sparsity-proximity trade-off.
>
> | Model | Classif. prob. ↑ | Proximity cont. ↓ | Sparsity cat. ↓ | ε-sparsity cont. ↓ | LOF log scale ↓ | Hypervol. log scale ↑ |
> |---|---|---|---|---|---|---|
> | DiCoFlex_p0.01 | **1.000** | **0.373** | 0.568 | **0.441** | **1.9119** | 0.9023 |
> | DiCoFlex_p0.08 | 0.996 | 0.501 | 0.553 | 0.454 | 1.9680 | 1.0801 |
> | DiCoFlex_p0.25 | 0.995 | 0.551 | 0.541 | 0.501 | 2.2380 | 1.3780 |
> | DiCoFlex_p1.0 | 0.996 | 0.577 | 0.537 | 0.589 | 2.6349 | **1.9491** |
> | DiCoFlex_p2.0 | 0.998 | 0.581 | **0.515** | 0.601 | 2.6614 | 1.8862 |
>
> The results reveal several key insights: Validity remains consistently high across all p values, confirming robust constraint satisfaction regardless of perturbation level. Clear proximity-sparsity trade-off: Lower p values achieve superior proximity and plausibility, while higher p values yield better categorical sparsity. Smooth transitions: All metrics change gradually across p values, enabling practitioners to fine-tune the balance between competing objectives based on their specific requirements.
>
> This sensitivity analysis validates our method's ability to provide flexible constraint handling, allowing users to systematically navigate the multi-objective trade-off space by simply adjusting the perturbation parameter p.
>
> We will include this analysis to our work in the final version.
>
> # Statistical Uncertainty Reporting
> We agree that statistical uncertainty measures enhance evaluation rigor. We calculate standard deviations across sets of diverse counterfactual explanations for each instance. We report standard deviation values for all individual-level metrics below. We will add standard deviation values to our main experiment results in Table 1 in the final version.
>
> | Dataset | Model | Validity ↑ | Classif. prob. ↑ | Proximity cont. ↓ | Sparsity cat. ↓ | ε-sparsity cont. ↓ | LOF log scale ↓ |
> |---|---|---|---|---|---|---|---|
> | Lending Club | DiCoFlex | 0 | 0.012 | 0.439 | 0.198 | 0.087 | 0.076 |
> | | s-DiCoFlex | 0 | 0.031 | 0.432 | 0.209 | 0.100 | 0.053 |
> | | CCHVAE | 0 | 0.368 | 0.401 | 0.049 | 0.066 | 0.119 |
> | | DiCE | 0 | 0.037 | 1.358 | 0.004 | 0.053 | 0.167 |
> | Adult | DiCoFlex | 0 | 0.011 | 0.518 | 0.099 | 0.127 | 0.256 |
> | | s-DiCoFlex | 0 | 0 | 0.352 | 0.106 | 0.096 | 0.812 |
> | | CCHVAE | 0 | 0.102 | 0.646 | 0.049 | 0.189 | 0.976 |
> | | DiCE | 0 | 0 | 1.464 | 0.017 | 0.264 | 0.915 |
> | Bank | DiCoFlex | 0 | 0.025 | 0.571 | 0.100 | 0.111 | 0.016 |
> | | s-DiCoFlex | 0 | 0.119 | 0.466 | 0.088 | 0.073 | 0.011 |
> | | CCHVAE | 0 | 0.053 | 0.606 | 0.061 | 0.090 | 0.078 |
> | | DiCE | 0 | 0.012 | 1.162 | 0.056 | 0.082 | 0.139 |
> | Default | DiCoFlex | 0 | 0.021 | 0.346 | 0.138 | 0.071 | 0.038 |
> | | s-DiCoFlex | 0 | 0.012 | 0.324 | 0.171 | 0.085 | 0.034 |
> | | CCHVAE | 0 | 0.043 | 0.273 | 0.097 | 0.076 | 0.047 |
> | | DiCE | 0 | 0.097 | 1.203 | 0.141 | 0.113 | 0.073 |
> | GMC | DiCoFlex | 0 | 0.032 | 0.535 | 0.235 | 0.058 | 0.061 |
> | | s-DiCoFlex | 0 | 0.037 | 0.694 | 0.229 | 0.059 | 0.099 |
> | | CCHVAE | 0 | 0.052 | 0.507 | 0.148 | 0.076 | 0.064 |
> | | DiCE | 0 | 0.072 | 1.021 | 0.048 | 0.073 | 0.019 |
>
> Note that we exclude Hypervolume from uncertainty reporting as this is a set-level metric that computes a single value per entire counterfactual set, making standard deviation calculations inappropriate.
>
> # Sparsity Explanation Correction
>
> Thank you for catching this error. The statement "p = 0.01 encourages lower sparsity" should read "p = 0.01 encourages higher sparsity" (fewer modified features). Lower p values in the Lp norm indeed encourage sparser solutions by penalizing modifications more heavily. We will correct this throughout the paper.
>
> # Context Vector Construction Details
> The conditioning vector concatenates: \[x (original instance), y (target class), p (sparsity parameter), m (actionability mask)\]. This vector is passed to each MAF layer through affine coupling transformations. This way we condition the estimated probability on the conditioning vector as it is described in [1]. We will add a detailed description of this architecture in the Appendix.
>
> [1] Papamakarios, George, Theo Pavlakou, and Iain Murray. "Masked autoregressive flow for density estimation." Advances in neural information processing systems 30 (2017).
>
> # Limitations
> Since DiCoFlex learns the counterfactual distribution for a specific classifier, classifier changes may invalidate some generated counterfactuals. However, generating multiple diverse counterfactuals allows filtering to retain valid ones, providing natural resilience without retraining.
>
> We believe these revisions will significantly strengthen the paper while maintaining its core contributions. Thank you again for your valuable feedback.

---

> > ### Comment · Reviewer_CeoU · 2025-08-02
> >
> > Thank you for answering my questiosn and providing additional information. I like this work, but I still think that in it's current shape, the comparison between DiCoFlex and s-DiCoFlex is somewhat confusing, therefore, I am hesitant to update my score upwards.

---

> > > ### Author Response · Authors · 2025-08-04
> > > **DiCoFlex Model Naming Concern**
> > >
> > > Thank you for your follow-up comment. We completely understand your confusion about DiCoFlex vs s-DiCoFlex, and we agree this naming might be misleading.
> > >
> > > We propose the following concrete changes for the camera-ready version:
> > > - **We will eliminate the separate "s-DiCoFlex" naming entirely** and present results as "DiCoFlex (p=2.0)" and "DiCoFlex (p=0.01)" throughout the paper, making it clear that these are configurations of the same model, not different methods.
> > >
> > > - **We will add the sensitivity analysis over p values table** (presented in our previous response) showing DiCoFlex's behavior across multiple p values. We will explicitly emphasize that this demonstrates a single model with the parameter p adjusted dynamically at inference time, requiring no retraining.
> > >
> > > - **We will add a clarifying statement in Section 4.1 (Parameterization of DiCoFlex):** "To demonstrate our method's flexible constraint control, we evaluate DiCoFlex with two different p values: p=2.0 (standard Euclidean distance) and p=0.01 (encouraging higher sparsity). These are not separate methods but rather showcase how users can dynamically adjust constraints at inference time without retraining."
> > >
> > > We believe these changes will transform what currently appears as a comparison between two methods into a clear demonstration of DiCoFlex's key strength: its ability to dynamically adjust constraints at inference time. This directly supports our core contribution of real-time user-driven customization and makes the paper's message much clearer.
> > >
> > > Would these concrete changes address your concern? We truly appreciate your feedback helping us identify this critical presentation issue.

---

### Official Review · Reviewer_TN7b · 2025-07-03

**Clarity:** 3
**Significance:** 3
**Originality:** 3
**Rating:** 4
**Confidence:** 4

**Summary:**

This paper introduces DiCoFlex, a model-agnostic, conditional generative framework for generating diverse counterfactual explanations for tabular data. DiCoFlex leverages conditional normalizing flows to generate multiple plausible counterfactuals in a single forward pass. The results demonstrate that DiCoFlex achieves significant computational efficiency and flexibility for incorporating domain-specific constraints.

**Questions:**

Please refer to the weakness

**Ethical Concerns:**

["NO or VERY MINOR ethics concerns only"]

**Final Justification:**

I have no questions about the submission. My only concern is that significant improvements made after the initial submission may require substantial revision.

**Limitations:**

Yes

**Quality:**

3

**Strengths And Weaknesses:**

## Strengths
1. The motivation is clear, and the proposed method is described in a detailed and structured way.
2. The experimental setup is thorough, with comprehensive evaluations across diverse datasets and meaningful comparisons to baseline methods.
3. The evaluation metrics for counterfactual quality, including diversity and plausibility provides extra perspectives.

## Weaknesses

1. There are gaps between the stated goals and the technical implementation. For example, the connection between validity, proximity, and plausibility and their calculation in Eq. (6) is not clearly articulated.
2. It is not clear how the score function (Eq. 4) and the actionable constraints Eq (9) are integrated into the training of the flow-based model.
3. The approach appears largely intuition-driven and lacks rigorous theoretical guarantees or proofs of properties such as diversity or constraint adherence.

---

> ### Author Rebuttal · Authors · 2025-07-30
>
> Thank you for your thoughtful review and constructive feedback. We appreciate your recognition of DiCoFlex's clear motivation, thorough experimental evaluation, and computational efficiency advantages.
>
> We address your main concerns below:
>
> # Clarification of Score Function Design (Eq. 6)
> We agree that introducing the score function given by equation (6) may be confusing and will clarify it in the revised version. This function is not used for training and evaluation. Our intention was to show how the constraints representing the validity, proximity, and plausibility can be aggregated into one criterion and to show that direct optimization of (6) can provide only a single counterfactual explanation, so it does not solve the main goal of this paper, which was focused on generating multiple counterfactuals efficiently.
>
> The connection between validity, proximity, and plausibility and their calculation in eq. (6) is as follows:
> logprob $\log p(y'|x')$ component represents the correct classification of conterfactual $x'$ to the desired class $y'$ (validity constraint).
> $d(x',x)$ represents the distance between the original example $x$, and the counterfactual $x'$ (with minus sign in equation (6), represents proximity constraint).
> $p_{data}(x')$ represents the density function for the data distribution (likelihood function representing plausibility constraint).
>
> As mentioned earlier, we use the score function only for motivation; direct optimization requires access to data distribution (can be modelled with flow) and leads to a counterfactual point, not the distribution of counterfactuals. Therefore, we propose to construct the conditional distribution for generating counterfactual $q(x'|x,y',d)$ given by eq. (7). The generated counterfactuals with this distribution satisfy constraints:
> Validity, because we select examples from a valid class $y'$.
> Proximity, because we take the nearest neighbours in class $y'$.
> Plausibility, because we take the real examples that exist in the data distribution.
>
> Besides, they also satisfy:
> Sparsity, because various norms can be used for distance calculation.
> Actionability, because selected attributes can be masked and remain unchanged.
>
> The unparametrized KNN-based model $q(x'|x,y',d)$ is further used to train a parametrized flow-based model $p_{\theta}(x'|x, y)$ by minimizing KLD given by eq. (1) that is equivalent to minimizing $Q$ given eq. (4). The technical implementation is described by the Algorithm in Appendix D (Supplementary Material).
>
> # Integration of Score Function and Constraints in Training
> Eq. (4) is the criterion used during training and is based on KLD between KNN-based model $q(x'|x,y',d)$  and flow-based model $p_{\theta}(x'|x, y',p,m)$ that is used to train parameters of the flow-based model. The technical procedure for training is described by the Algorithm in Appendix D. In each iteration, we select an example $x$ from the training data and sample a class $y'$. We sample the sparsity level $p$ and mask the candidate $m$. Next, we sample $x'$, one of the $K$ nearest neighbours of $x$ in class $y'$. Note that neighbors are calculated using the distance given by the eq. (9) based on the current $p$ and $m$ values. Finally, the parameters of the flow-based model $p_{\theta}(x'|x, y',p,m)$ are updated with a gradient-based procedure by minimizing $-\log(p_{\theta}(x'|x, y',p,m))$ (based on eq. (4)). We highlight, that we use condition variant of flow-based model, that takes $x$, $y'$ and sparsity parameter $s$ and actionability parametr represetned by mask $m$. So the flow-based model is trained by optimizing Eq. (4) using the neighbours calculated with the distance given by Eq. (9) with $m$ and $p$ sampled in each iteration. Because we sample $m$ and $p$  in each iteration during the training flow-based model, while generating counterfactuals, we can use this model to control sparsity and actionability via simple conditioning on $m$ and $p$.
>
> # Theoretical Considerations
>
> We disagree that our approach is largely intuition-driven and lacks rigorous theoretical guarantees. We aim to construct a distribution $q(x'|x,y',d)$ for which the samples represent counterfactuals that satisfy validity (examples from correct class y'), proximity (the closest examples, because we use KNN), plausibility (we select examples that are in data distribution), sparsity (we control the norm with parameter $p$), and actionability (we selected only attributes that can be changed using masking procedure). In lines 175-181, we justify not taking $q(x'|x,y',d)$ directly as a sampling model. Finally, we formulate the theoretical problem of finding a parametrized probabilistic model as a surrogate model in eq. (1), and show how to solve it efficiently.
>
> Additionally, we provide a theorem for theoretical guarantees of diversity preservation for our method that will be included into the final version of our work:
>
> ## Lemma: Upper bound on difference of expected values between two distributions
>
> Let $p, q$ be two distributions over the same finite set $X$ and the total variation distance $TV(p,q) = \frac{1}{2}\sum_x |p(x) - q(x)|$. If for some function $f: X \to \mathbb{R}$, then:
>
> $$\left|\sum_x f(x)(p(x) - q(x))\right| \leq \max_x |f(x)| \cdot \sum_x |p(x) - q(x)| = 2 \cdot \max_x |f(x)| \cdot TV(p,q)$$
>
> ## Theorem: Diversity Preservation in DiCoFlex
> ### Statement
> Let $p_\omega(x'|x,y',p,m)$ be the learned conditional normalizing flow and $\hat{q}(x'|x,y',d_{p,m})$ be the empirical distribution defined in Eq. (7). Under assumptions (A1)-(A4), for n samples $x_i', i = 1, 2, \ldots, n$ drawn from $p_\omega$, the expected diversity satisfies:
>
> $$
> \mathbb{E}\_{p\_\omega}[D(\{x'\_i\}\_{i=1}^n)] \geq \sigma\_{\min}(1 - K^{1-n}) - \Delta\sqrt{2\epsilon}
> $$
>
> ### Assumptions
> - **(A1)** The neighborhood $N(x, y', d_{p,m}, K)$ contains $K$ distinct points with minimum pairwise distance $\sigma_{\min} > 0$. Specifically, for any $x_i, x_j \in N(x, y', d_{p,m}, K)$ with $i \neq j$,
> $$
> |x_i - x_j| \geq \sigma_{\min}
> $$
> - **(A2)** The normalizing flow is trained sufficiently close to the empirical distribution, satisfying:
> $$
> D_{KL}(\hat{q}(x'|x,y',d_{p,m}) \| p_\omega(x'|x,y',p,m)) \leq \epsilon
> $$
> for all $(x,y',p,m)$ in the training distribution.
> - **(A3)** The support of $p_\omega(\cdot\mid x,y',p,m)$ lies in a set of finite diameter $\Delta$, i.e.
>
> $$
> |x' - x''| \leq \Delta \quad \Longrightarrow \quad 0 \leq D(\{x'\_i\}) = \min\_{i \neq j} |x'_i - x'_j| \leq \Delta.
> $$
>
> ### Proof
> **Step 1: Empirical Diversity**
> Consider the empirical distribution $\hat{q}$. When sampling $n$ points uniformly at random (with replacement) from $K$ distinct points, the expected diversity is at least the product of minimum pairwise distance $\sigma_{\min}$ and the probability of selecting at least two distinct points $P_n(K)$:
>
> $$
> \mathbb{E}\_{\hat{q}}[D(\{x'\_i\}_{i=1}^n)] \geq \sigma\_{\min} \cdot P_n(K) = \sigma\_{\min}(1 - K^{1-n})
> $$
>
> **Step 2: Total Variation Distance via Pinsker’s Inequality**
> Pinsker's inequality relates the KL-divergence and total variation distance:
>
> $$
> ||p\_\omega(\cdot|x,y',p,m) - \hat{q}(\cdot|x,y',d\_{p,m})||\_{TV} \leq \sqrt{\frac{\epsilon}{2}}
> $$
>
> **Step 3: Bounding the Difference in Diversity**
> The diversity measure, defined by pairwise distances, is bounded above by $\Delta$. Thus, using Lemma 1 we obtain:
>
> $$
> |\mathbb{E}\_{p\_\omega}[D] - \mathbb{E}\_{\hat{q}}[D]| \leq 2 \cdot \Delta \cdot ||p\_\omega - \hat{q}||\_{TV} \leq 2 \Delta\sqrt{\frac{\epsilon}{2}} = \Delta\sqrt{2\epsilon}
> $$
>
> **Step 4: Combining the Results**
> From Step 3, we have an absolute value bound. Since we specifically seek a lower bound for $\mathbb{E}\_{p_\omega}[D]$, we remove the absolute value by explicitly considering the worst-case scenario for diversity under $p\_\omega$ relative to $\hat{q}$:
>
> $$
> \mathbb{E}\_{p_\omega}[D(\{x'_i\}\_{i=1}^n)] \geq \mathbb{E}\_{\hat{q}}[D(\{x'_i\}\_{i=1}^n)] - \Delta\sqrt{2\epsilon}
> $$
>
> Substituting Step 1 into the above, we obtain the desired explicit lower bound:
>
> $$
> \mathbb{E}\_{p\_\omega}[D(\{x'_i\}\_{i=1}^n)] \geq \sigma\_{\min}(1 - K^{1-n}) - \Delta\sqrt{2\epsilon}
> $$
>
> ### Remarks
> 1. **Diversity Guarantee**: For large $K$, $1 - K^{1-n}\approx 1$, ensuring strong diversity. Thus the bound
>
> $$
> \mathbb{E}\_{p\_\omega}[D(\{x'_i\}\_{i=1}^n)] \geq \sigma\_{\min}(1 - K^{1-n}) - \Delta\sqrt{2\epsilon}
> $$
>
> guarantees that $p\_\omega$ recovers nearly the full empirical diversity except for the training error.
>
> 2. **Impact of Training**: As training improves ($\epsilon \rightarrow 0$), the perturbation term $\Delta\sqrt{2\epsilon}$ vanishes, and the learned distribution’s diversity converges to the empirical distribution’s diversity.

---

> > ### Comment · Reviewer_TN7b · 2025-08-04
> > **thanks for the reply**
> >
> > Dear authors,
> >
> > Thanks for your detailed replies.
> >
> > Two quick follow-up clarifications:
> > 1. Does this paper propose Eq 7 as an alternative and intuitive solution to the challenging direct optimization (Eq 6.)?
> > 2. The new added theoretical guarantees is for diversity only. Is there any theoretical guarantee for validity, proximity, and/or plausibility of Eq 7?
> >
> > Thanks

---

> > > ### Author Response · Authors · 2025-08-05
> > >
> > > Thank you for these insightful follow-up questions. We will now clarify the relationship between Equations 6 and 7, and provide the theoretical guarantees you've requested.
> > >
> > > # Relationship between Eq. 6 and Eq. 7:
> > >
> > > Eq. 7 is not proposed as an alternative solution to Eq. 6. Rather, Eq. 6 serves only as motivation to illustrate how validity, proximity, and plausibility constraints could theoretically be aggregated into a single score function. We discuss that direct optimization of Eq. 6 would only produce a single counterfactual, which fails to address our main goal of generating multiple diverse counterfactuals efficiently.
> > >
> > > Eq. 7 represents our actual proposed approach: a KNN-based distribution $\hat{q}(x'|x, y', d)$ that inherently satisfies the desired constraints (validity by selecting from class y', proximity through nearest neighbors, plausibility by using real data examples, plus sparsity and actionability through distance metric choice). This distribution is not meant to replace Eq. 6, but rather serves as the training data source for learning the parametrized flow model $p_\theta(x'|x,y')$.
> > >
> > > # Theoretical Guarantees for Eq. 7:
> > >
> > > ## Validity
> > >
> > > Our method provides a validity guarantee by definition. Since $\hat{q}(x'|x, y', d)$ (Eq. 7) only samples from $N(x, y', d, K)$ - the set of K-nearest neighbors where $h(x_i) = y'$ - we have **$p(y'|x') = 1$ for all training samples,** where p is class probability returned by classification model.
> > >
> > > This theoretical property explains why DiCoFlex achieves 100% validity across all datasets while being orders of magnitude faster than optimization-based methods. The normalizing flow trained on these valid samples inherits this property in expectation: $\mathbb{E}\_{x' \sim p\_\theta}[p(y'|x')] \approx 1$.
> > >
> > > ## Proximity
> > >
> > > Our K-NN sampling mechanism provides an implicit theoretical guarantee on proximity. For any counterfactual x' sampled from $\hat{q}(x'|x, y', d)$ (Eq. 7), we have:
> > >
> > > $$d(x, x') \leq d(x, x_K^{y'})$$
> > >
> > > where $x_K^{y'}$ is the K-th nearest neighbor of x in class y'.
> > >
> > > **Proof:** By definition of $\hat{q}$, we only sample from $N(x, y', d, K)$. Any x' outside this set has $\hat{q}(x'|x, y', d) = 0$. Therefore, d(x, x') cannot exceed the distance to the K-th nearest neighbor, as otherwise x' would not be in $N(x, y', d, K)$.
> > > This provides an upper bound on proximity without requiring explicit proximity optimization.
> > >
> > > ## Plausibility
> > >
> > > We can indeed provide theoretical guarantees for plausibility. Since our training distribution $\hat{q}(x'|x, y', d)$ only assigns non-zero probability to K-nearest neighbors from the training set (Eq. 7), any sampled counterfactual $x'$ satisfies:
> > >
> > > $$p_{data}(x') \geq \min_{x_i \in \mathcal{D}} p_{data}(x_i) > 0$$
> > >
> > > This guarantees that training samples are in-distribution. Furthermore, when using small $p$ values (e.g., 0.01) in the $L_p$ norm, we restrict to neighbors with minimal feature changes, which empirically correspond to higher-density regions, explaining the improved LOF scores for s-DiCoFlex in Table 1.
> > >
> > > The normalizing flow then learns to interpolate between these guaranteed in-distribution points while preserving the manifold structure. This provides a principled approach ensuring plausibility, addressing a key limitation of optimization-based methods that may generate out-of-distribution counterfactuals.

---

### Note · Authors · 2025-08-13

We sincerely thank all reviewers for their constructive and insightful feedback, which has led to a productive discussion and improvements to our work.

Reviewers acknowledged our key contributions: model-agnostic approach with dynamic inference-time constraint control (CeoU), novel use of normalizing flows to model counterfactual desiderata including actionability (jqT4), significant computational efficiency without quality compromise (all reviewers), thorough experimental evaluation across diverse datasets (TN7b, CeoU), and high reproducibility with provided code (CeoU).

Reviewer TN7b: We strengthened theoretical foundations with formal diversity preservation guarantees (theorem with proof) and theoretical guarantees for validity, proximity, and plausibility. We clarified Eq. 6's role and why direct optimization yields single counterfactuals, highlighting our generative approach's advantage. The reviewer confirmed these additions address all theoretical concerns.

Reviewer CeoU: We successfully resolved the confusion regarding DiCoFlex vs. s-DiCoFlex naming, confirming these are evaluation configurations of the same model (DiCoFlex with p=2.0 and p=0.01) demonstrating dynamic constraint control without retraining. We provided comprehensive sensitivity analysis across $p\in\\{0.01,0.08,0.25,1.0,2.0\\}$, showing smooth control over sparsity-proximity trade-offs. We added statistical uncertainty reporting for all metrics and clarified the context vector construction details. The reviewer confirmed our proposed changes address their concerns.

Reviewer jqT4: We addressed all technical concerns by adding experiments on smaller datasets (German Credit with 1,000 samples), demonstrating strong generalizability. We provided detailed training time analysis and clarified the counterfactual generation process through sampling from the learned normalizing flow. The reviewer explicitly acknowledged our rebuttal addressed their concerns and improved score accordingly.

All reviewers engaged positively with our responses throughout the discussion phase. We successfully resolved additional questions that emerged during follow-up exchanges. Reviewer jqT4 explicitly raised their score after the rebuttal. These productive discussions resulted in a strengthened paper that maintains our core contributions while addressing all concerns. We are committed to incorporating all discussed improvements in the camera-ready version.

Thank you for this valuable rebuttal process.

---

### Decision · Program_Chairs · 2025-09-17

**Decision:**

Accept (poster)

**Comment:**

The paper introduces DiCoFlex, a novel model-agnostic framework for generating diverse counterfactual explanations using conditional normalizing flows. All reviewers acknowledged the method's key innovation of enabling dynamic constraint control at inference time without retraining, achieving orders of magnitude faster generation than optimization-based methods while maintaining or improving quality metrics. The approach was validated across 5 tabular datasets with comprehensive metrics including validity, proximity, plausibility, and diversity, with code provided for reproducibility.

Initially, reviewers raised concerns about theoretical foundations (TN7b), method clarity and DiCoFlex vs. s-DiCoFlex naming confusion (CeoU, jqT4), missing training details and evaluation on smaller datasets (jqT4), and lack of statistical uncertainty reporting (CeoU). All reviewers who engaged post-rebuttal were satisfied with the responses, with jqT4 explicitly raising their score after additional experiments on German Credit dataset were provided, and TN7b confirming that theoretical guarantees with formal proofs addressed all concerns.

While this is a solid technical contribution with practical advantages in computational efficiency and flexible constraint handling, it represents an incremental advance in counterfactual explanation methods rather than a breakthrough result. It is good to present in poster.